# Phosphodiesterase 2A2 regulates mitochondria clearance through Parkin-dependent mitophagy

Miguel J. Lobo[1], Laia Reverte-Salisa[2], Ying-Chi Chao[1], Andreas Koschinski[1], Frank Gesellchen[3], Gunasekaran Subramaniam[1], He Jiang[3], Samuel Pace[1], Natasha Larcom[1], Ester Paolocci[1], Alexander Pfeifer [2], Sara Zanivan [4,5] & Manuela Zaccolo [1✉]

Programmed degradation of mitochondria by mitophagy, an essential process to maintain mitochondrial homeostasis, is not completely understood. Here we uncover a regulatory process that controls mitophagy and involves the cAMP-degrading enzyme phosphodiesterase 2A2 (PDE2A2). We find that PDE2A2 is part of a mitochondrial signalosome at the mitochondrial inner membrane where it interacts with the mitochondrial contact site and organizing system (MICOS). As part of this compartmentalised signalling system PDE2A2 regulates PKA-mediated phosphorylation of the MICOS component MIC60, resulting in modulation of Parkin recruitment to the mitochondria and mitophagy. Inhibition of PDE2A2 is sufficient to regulate mitophagy in the absence of other triggers, highlighting the physiological relevance of PDE2A2 in this process. Pharmacological inhibition of PDE2 promotes a 'fat-burning' phenotype to retain thermogenic beige adipocytes, indicating that PDE2A2 may serve as a novel target with potential for developing therapies for metabolic disorders.

[1] Department of Physiology, Anatomy and Genetics, University of Oxford, Oxford, UK. [2] Institute of Pharmacology and Toxicology University of Bonn, Bonn, Germany. [3] Institute of Neuroscience and Psychology, University of Glasgow, Glasgow, UK. [4] Cancer Research UK Beatson Institute, University of Glasgow, Glasgow, UK. [5] Institute of Cancer Sciences, University of Glasgow, Glasgow, UK. ✉email: manuela.zaccolo@dpag.ox.ac.uk

# 3′-5′-c

yclic adenosine monophosphate (cAMP) is a ubiquitous and pleiotropic second messenger. Among its many functions, cAMP mediates the response to multiple neurotransmitters in the nervous system, is the key controller of rate and strength of cardiac contraction and is a master regulator of cellular metabolism. Synthesis of cAMP is triggered by stimulation of G protein-coupled receptors and activation of adenylyl cyclase (AC) enzymes anchored at the plasma membrane[1] or activation of an intracellular AC sensitive to bicarbonate[2]. A rise in intracellular cAMP can translate, within the same cell, into a multitude of different functional effects, from modulation of metabolism to regulation of gene transcription, from control of cell division to regulation of cell death. The complex and in some case opposing effects of cAMP require precise coordination in order for a specific extracellular stimulus to result in the appropriate cellular response, a process that relies on compartmentalisation of the signaling events at distinct signalosomes[3]. Typically, within a signalosome, an A Kinase anchoring protein (AKAP) segregates the main cAMP effector, the protein kinase A (PKA), in proximity to a specific phosphorylation target[4]. Selective activation of differently localized PKA subsets is achieved by confining the increase in cAMP triggered by a specific stimulus to the subcellular site, or nanodomain, within which the relevant AKAP/PKA/target signalosome resides, enabling rapid and selective PKA-dependent phosphorylation[5–7]. Phosphodiesterases (PDEs), a superfamily of enzymes that degrade cyclic nucleotides and comprise 11 families (PDE1-11) with multiple isoforms, play a central role in confining cAMP within individual subcellular nanodomains and in determining the amplitude and kinetics of the local cAMP signal[8]. This is often achieved via anchoring specific PDE isoforms to the signalosome via protein-protein interactions[9].

Mitochondria are indispensable organelles with central functions in cellular energetics, metabolism and survival. Dysfunctional mitochondria have been linked to pathological conditions ranging from neurodegeneration and cardiomyopathies to cancer and metabolic disorders[10]. Consistent with their multifunctional role, mitochondria are under the control of a complex system of signaling pathways and regulatory mechanisms. They form a highly dynamic network where fusion and fission events and quality control processes are fine-tuned to maintain cellular energy homeostasis[11,12]. The cAMP/PKA signaling pathway has emerged as a direct means to regulate mitochondria physiology[13–16] via PKA-dependent phosphorylation of several mitochondrial proteins involved in mitophagy and fusion/fission dynamics[17–19]. An example of cAMP-dependent regulation of mitochondrial dynamics is adipocyte plasticity. In adipocytes, cAMP is pivotal in early stages of differentiation and in determining the transition between white adipocytes, that store energy in the form of fat, and beige adipocytes, that dissipate energy by burning lipids. Beige adipocytes mimic brown adipocytes with multiple small lipid droplets and high content of mitochondria that express the uncoupling protein 1 (UCP1)[20]. This process is known as 'browning' or 'beiging' and is characterized by reduced mitophagy and accumulation of mitochondria[21–23].

In mitochondria, multiple AKAPs anchor PKA to different organelle compartments. For example, AKAP1 docks PKA to the mitochondrial outer membrane (MOM)[24] whereas the AKAP sphingosine kinase interacting protein (SKIP) tethers PKA to the mitochondria intermembrane space (MIS)[25]. The presence of several AKAPs at the mitochondria suggests that multiple and functionally independent cAMP/PKA signaling domains may operate at this organelle[26]. However, the specific functional role of these distinct submitochondrial domains and their regulation is not completely understood.

PDE2A, a dual-specificity PDE family, is expressed in a number of tissues including brain, heart, liver, lung, adrenal gland, and adipose tissue[27]. PDE2A contains a tandem pair of regulatory domains (GAF-A and GAF-B). While the function of GAF-A is unclear, binding of cGMP to the GAF-B domain allosterically activates PDE2A catalytic activity[28]. Genetic ablation of *Pde2a* results in embryonic lethality[29], indicating its involvement in fundamental biological functions. Three protein variants of PDE2A are expressed (PDE2A1, PDE2A2, and PDE2A3) that differ in their amino termini, which are responsible for different subcellular localisation[30]. Of these isoforms, PDE2A2 localizes to the mitochondria, PDE2A1 is predominantly cytosolic and PDE2A3 localizes largely to the plasma membrane[31]. At the mitochondria, a subset of PDE2A2 was reported to reside in the matrix and to control oxidative phosphorylation[32]. In previous studies, we found that PDE2A2 localizes to mitochondrial membranes, largely at the mitochondrial inner membrane (MIM), and regulates mitochondria fusion/fission[31]. A fraction of PDE2A2 localizes outside the mitochondria[31].

In an attempt to define the molecular basis for the differential subcellular localization of PDE2A2, here we studied the PDE2A2 interactome using mass spectrometry (MS). Our analysis reveals that PDE2A2 interacts with the mitochondrial contact site and organizing system (MICOS) localized at the MIM. We show that, in a variety of cell lines and primary cells, PDE2A2 regulates a local pool of cAMP that controls PKA-dependent phosphorylation of the MICOS component MIC60. We further demonstrate that modulation of PDE2A2 activity at MICOS regulates recruitment of Parkin to the organelle and mitophagy and that PDE2A2 inhibition promotes adipocyte browning.

## Results

**PDE2A2 interacts with MICOS.** To determine the interactome of PDE2A2, we used MS-based proteomics. For this purpose, Strep-tagged PDE2A2 and, as a control, Strep-tagged elongation factor thermo stable (EF-Ts) were overexpressed in the mouse neuroblastoma cell line HT-4 and immunoprecipitated using an anti-Strep-tag antibody. The pulldown proteins were analyzed by MS and accurately quantified using the LFQ algorithm of the MaxQuant computational platform[33]. A full list of interactors that we found to be significantly enriched in the PDE2A2 pull down is shown in Supplementary Data 1. Consistent with previous evidence[31,32,34], the MS screen revealed a large number of mitochondrial proteins (Fig. 1a). Gene Ontology Cellular Compartment (GOCC) term analysis (Supplementary Data 2) showed a significant enrichment selectively for proteins localized at mitochondrial membranes (term "Mitochondrial membrane", *p*-value 3.13E-05 and FDR 2.61E-3). Among those, we identified two core components of MICOS, namely MIC60 (IMMT) and MIC19 (ChChD3), as well as SAMM50 and Metaxin-2 (MTX-2), a core subunit and an associate component, respectively, of the Sorting and Assembly machinery (SAM), a protein complex known to interact with MICOS[35–38] (Fig. 1a).

Consistent with an interaction of PDE2A2 with MICOS, immunolabelling of the neuronal cell line ShSy5y expressing PDE2A2-GFP with an antibody specific for MIC60 shows signal overlap at the mitochondria (Fig. 1b). Interaction of PDE2A2 with MICOS was confirmed by co-immunoprecipitation from HEK293T cells expressing PDE2A2-RFP and MIC60-FLAG, SAMM50-FLAG or Metaxin2-FLAG (Supplementary Fig. 1a–d). Interaction of PDE2A2 with MICOS was also confirmed by co-immunoprecipitation of endogenous MIC60 (Fig. 1c), MIC19 (Fig. 1d) or SAMM50 (Fig. 1e) from HEK293T cells expressing

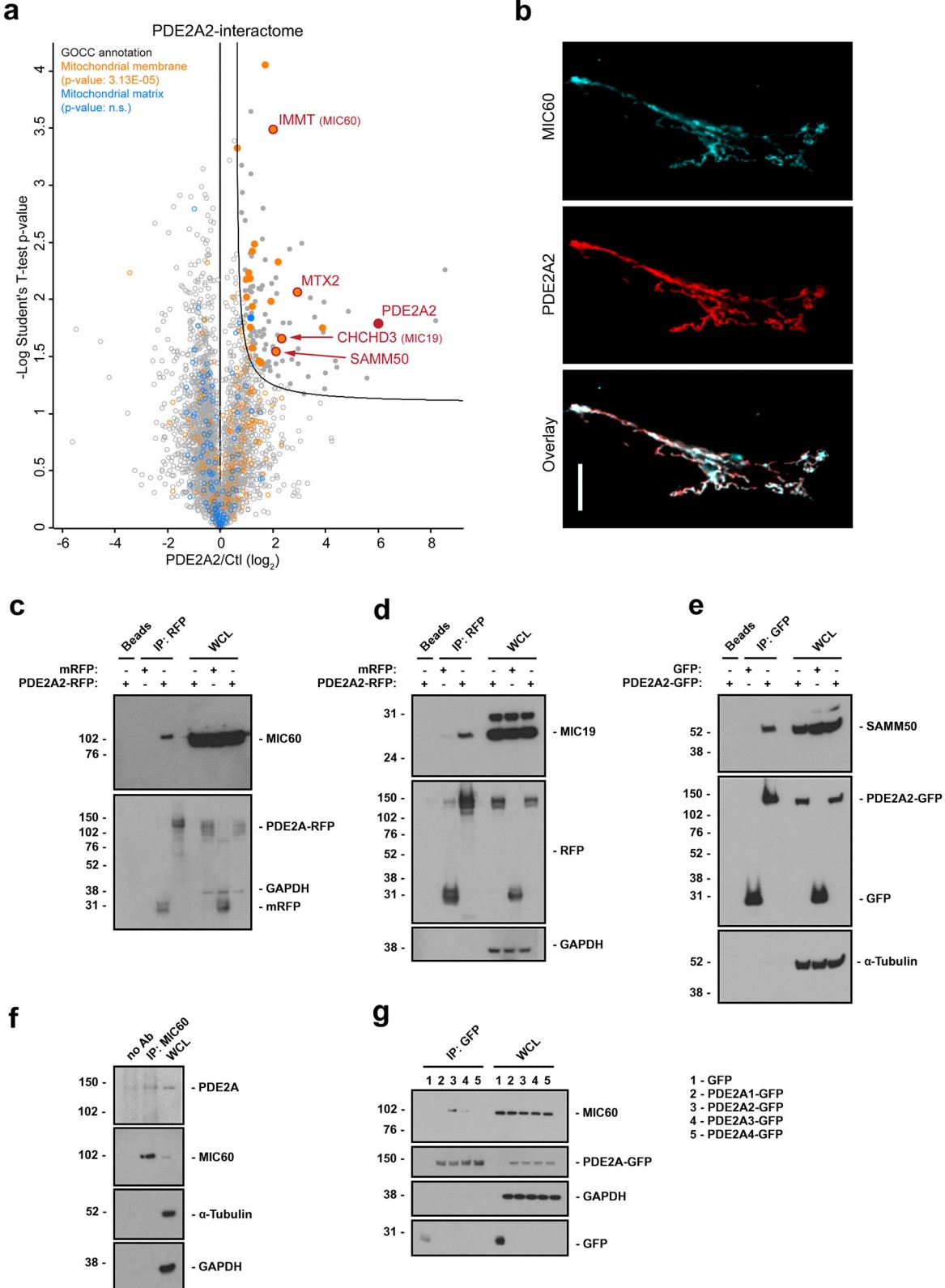

PDE2A2-RFP. In addition, endogenous PDE2A was detected in the immunoprecipitate from rat brain homogenates obtained by pulldown of endogenous MIC60 (Fig. 1f). The interaction of MICOS with PDE2A is isoform-specific as co-immunoprecipitation experiments in HEK293T cell expressing PDE2A1-GFP, PDE2A2-GFP or PDE2A3-GFP and MIC60-FLAG show selective interaction of MIC60-FLAG with PDE2A2-GFP (Fig. 1g).

**PDE2A2 interacts with MICOS through the GAF-B domain**. To further characterize the interaction between PDE2A2 and MICOS we generated a set of PDE2A2-RFP deletion mutants, as illustrated in Supplementary Fig. 2a, b. The N-terminus is considered to be the region upstream (1-201 aa) of the GAF-A and GAF-B domains. Consistent with previous findings[32], confocal images of HeLa cells transiently expressing the PDE2A2-RFP

**Fig. 1 PDE2A2 interacts with MICOS components. a** Volcano plot of the PDE2A2 interactome. Filled circles indicate proteins significantly pulled-down with PDE2A2. Highlighted in orange are proteins annotated to the GOCC category "Mitochondrial membrane" and in blue are proteins annotated to the category 'Mitochondrial matrix'. PDE2A2, MICOS components and MICOS-interacting proteins are shown in red. **b** Confocal images of Sh-Sy5y cells expressing PDE2A2-GFP. Cells were fixed and immunostained with GFP- (red), MIC60- (turquoise) specific antibodies. Images are representative of two independent experiments. Scale bar: 10 μm. **c** Detection of endogenous MIC60 after PDE2A2-RFP or mRFP immunoprecipitation from lysates obtained from HEK293T cells transiently expressing either PDE2A2-RFP or mRFP. **d** Detection of endogenous MIC19 after PDE2A2-RFP or mRFP immunoprecipitation from lysates obtained from HEK293T cells transiently expressing either PDE2A2-RFP or mRFP. **e** Detection of endogenous SAMM50 after PDE2A2-GFP or GFP immunoprecipitation from lysates obtained from HEK293T cells transiently expressing either PDE2A2-GFP or GFP. **f** Detection of endogenous PDE2A after immunoprecipitation of endogenous MIC60 from rat brain cell lysates. **g** Detection of endogenous MIC60 after immunoprecipitation of each PDE2A isoform from lysates obtained from HEK293T cells transiently expressing PDE2A1-GFP, PDE2A2-GFP, PDE2A3-GFP or PDE2A4-GFP. PDE2A4, a variant whose sequence has been deposited (NCBI Reference Sequence: NM_001146209.2) but for which no further characterization is available, was used here as a control. All blots are representative of three independent experiments except from data shown in (**f**), which is representative of two independent experiments. WCL whole-cell lysate.

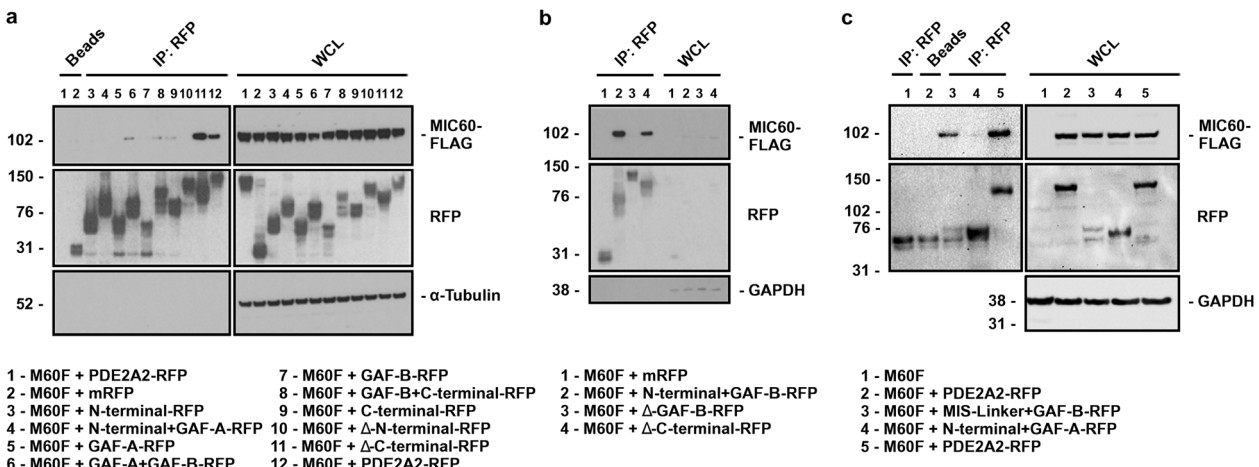

**Fig. 2 The GAF-B domain of PDE2A2 is necessary for the interaction with MIC60. a** Detection of MIC60-FLAG after PDE2A2-RFP fragments immunoprecipitation from cell lysates obtained from HEK293T cells transiently expressing PDE2A2-RFP deletion constructs and MIC60-FLAG. Pull-down of mRFP was used as a negative control. Blot is representative of three independent experiments. **b** Detection of MIC60-FLAG after immunoprecipitation of PDE2A2-RFP fragments from cell lysates obtained from HEK293T cells transiently expressing PDE2A2-RFP constructs and MIC60-FLAG, as indicated. Pull-down of mRFP was used as a negative control. Blot is representative of two independent experiments. **c** Detection of MIC60-FLAG after immunoprecipitation of N-terminal+GAF-B-RFP or MIS-linker+GAF-B-RFP from cell lysates obtained from HEK293T cells transiently expressing the constructs. Pull-down from untransfected cell lysate was used as a negative control. Blot is representative of three independent experiments.

fragments (Supplementary Fig. 2c) and Western blot analysis of subcellular mitochondrial fractions generated from HEK293T expressing the PDE2A2-RFP deletion mutants (Supplementary Fig. 2d), confirmed that only fragments including the N-terminal region localize to the mitochondria. Further dissection of the PDE2A2 N-terminal domain (Supplementary Fig. 3b) revealed that, although required, the unique PDE2A2 region spanning amino acid 1-17 is not sufficient to target the protein to the mitochondria and that additional residues between aa 17 and aa 201 are required for full targeting (Supplementary Fig. 3b, c).

To characterize the interaction between MICOS and PDE2A2, co-immunoprecipitation experiments were conducted in HEK293T cells expressing PDE2A2-RFP deletion mutants and MIC60-FLAG. As shown in Fig. 2a, only fragments that contain the GAF-B domain and are targeted to the mitochondria (i.e. included the PDE2A2 N-terminal domain) co-immunoprecipitated with MIC60. Consistently, two additional deletion mutants, N-terminal+GAF-B-RFP and ΔGAF-B-RFP (Supplementary Fig. 3d), both of which localise to the mitochondria as expected (Supplementary Fig. 3e, f), co-immunoprecipitate with MIC60, confirming that the GAF-B domain of PDE2A2 contains the interaction site for MICOS although, for this interaction to occur, the PDE2A2 N-terminus is required to target the GAF-B domain to the mitochondria (Fig. 2b). To further confirm that the GAF-B domain, when

targeted to the mitochondria, is sufficient for binding to MICOS, we generated a modified version of N-terminal+GAF-B-RFP where we substituted the PDE2A2 mitochondrial targeting domain (1-41 aa) with the mitochondrial targeting sequence from the protein Smac/DIABLO[39] (1-59 aa) (Supplementary Fig. 3d). We found that this chimera correctly localises to the mitochondria (Supplementary Fig. 3e, f) and efficiently interact with MIC60 (Fig. 2c). The fact that all PDE2A isoforms contain the GAF-B domain (Supplementary Fig. 3a) but only PDE2A2 interacts with MICOS indicates that this interaction only occurs at the mitochondria.

**PDE2A2 regulates PKA-mediated phosphorylation of MIC60.** MIC60 is phosphorylated by PKA[19]. Consistently, we found that HEK293T cells expressing MIC60-FLAG and treated with the AC activator forskolin (1 μM) for 2 h show significantly increased PKA-dependent phosphorylation of MIC60 compared to control (Fig. 3a). Given the interaction of PDE2A2 with MICOS, we hypothesized that PDE2A2 hydrolyzes a local pool of cAMP at the mitochondria that regulates PKA-dependent phosphorylation of MIC60. To test this hypothesis, we treated HEK293 cells with the PDE2A selective inhibitor Bay 60-7550 (BAY) for 2 h and found significant enhancement of MIC60 phosphorylation compared to control (Fig. 3a, longer exposure panel and 3b). The increased level of MIC60 phosphorylation observed is PKA-

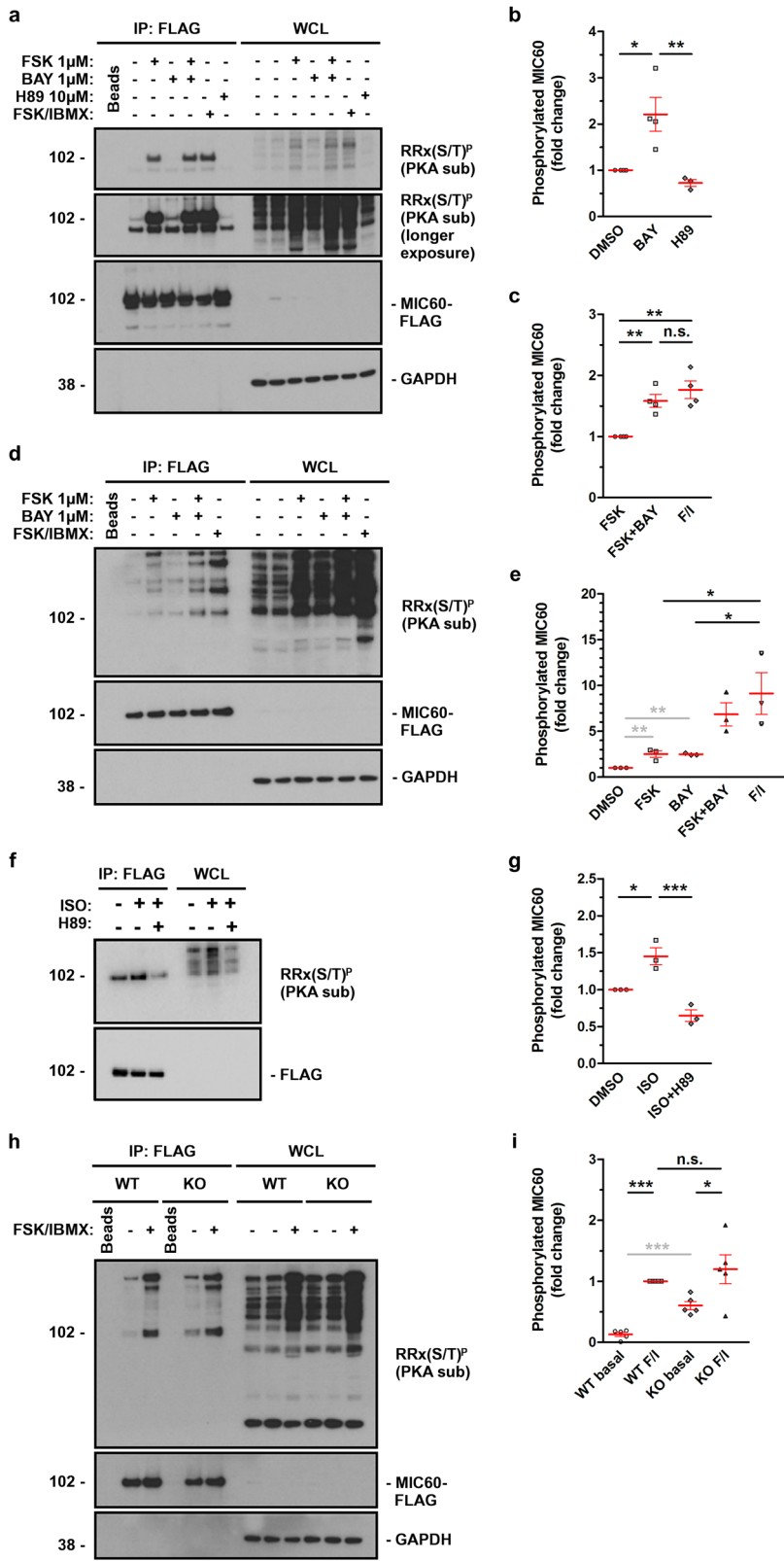

dependent, as it was blocked by treatment with the PKA inhibitor H89 (Supplementary Fig. 4a, b). Combined application of BAY and Forskolin showed an additive effect that was not significantly different from treatment with forskolin in combination with the global PDE inhibitor Isobutyl-methyl-xanthine (IBMX) (Fig. 3a, c), suggesting that PDE2A is the predominant PDE responsible for the modulation of PKA-mediated phosphorylation of MIC60.

Similar results were observed in the human neuroblastoma cell line Sh-Sy5y (Fig. 3d, e). Increased PKA-dependent phosphorylation of MIC60 was found also when cAMP synthesis in HEK293 was induced via a physiological activation of β-adrenergic receptors with low doses (10 nM) of Isoproterenol (ISO) (Fig. 3f, g). We also examined the level of PKA-dependent phosphorylation of MIC60 in mouse embryonic fibroblasts

**Fig. 3 PDE2A controls the phosphorylation status of MIC60. a** Representative western blot and (**b, c**) quantification of the phosphorylation status of MIC60 in HEK293T cells expressing MIC60-FLAG and treated with Forskolin for 2 h. $N = 4$. **d** Representative western blot and **e** quantification of MIC60 phosphorylation status in ShSy5y cells overexpressing MIC60-FLAG and treated with Forskolin for 2 h. $N = 3$. **f** Representative western blot and **g** quantification of MIC60 phosphorylation level detected in HEK293T cells expressing MIC60-FLAG and treated with ISO (10 nM) alone or in combination with H89 for 2 h. $N = 3$. **h** Representative western blot and **i** quantification of MIC60 phosphorylation in WT or PDE2A$^{-/-}$ MEF cells expressing MIC60-FLAG and either treated or non-treated with Forskolin and IBMX for 1 h. $N = 5$. For all panels, a phospho-(Ser/Thr) PKA substrate antibody (RRx(S/T)$^P$) was used both on whole-cell lysate (WCL) and on MIC60-FLAG pulled-down samples (IP). The amount of phosphorylated MIC60 was normalized to the amount of precipitated MIC60 and is shown as the fold change relative to levels in DMSO-treated cells, except for (**c**) and (**i**), where values are shown as the fold change relative to levels in Forskolin-treated cells and as the fold change relative to levels detected in WT cells treated with Forskolin/IBMX (FI), respectively. All values represent mean ± s.e.m. One-way ANOVA and Tukey's multiple comparison tests. Student's $t$-test comparisons are shown in gray *$P < 0.05$; **$P < 0.01$; ***$P < 0.005$; ns = not significant.

obtained from a line homozygous for the deletion of the PDE2A gene (MEF$^{PDE2A-/-}$)[31] and found a higher baseline level of MIC60 phosphorylation compared to wild type cells (MEF$^{WT}$) (Fig. 3h, i). Consistently, overexpression of PDE2A2 completely abolished the increased phosphorylation of MIC60 resulting from treatment with 1 μM forskolin (Supplementary Fig. 4c, d). Overall, the above data demonstrate that PDE2A2 regulates PKA-dependent phosphorylation of MIC60.

**PDE2A2 promotes recruitment of Parkin and mitophagy.** PKA-dependent phosphorylation of MIC60 inhibits recruitment of Parkin to the MOM[19]. We therefore investigated whether MICOS-interacting PDE2A2 may regulate this process. When cells expressing mCherry-Parkin were treated with the protonophore carbonyl cyanide m-chlorophenyl hydrazone (CCCP), Parkin was recruited to mitochondria (Supplementary Fig. 5). In line with previous findings[19,40], the elevation of cAMP in MEF$^{WT}$ treated with Forskolin significantly reduced, in a concentration-dependent manner, mitochondrial recruitment of Parkin (Fig. 4a). To assess a possible regulatory role of PDE2A2 on mitochondrial Parkin recruitment we expressed in MEF$^{WT}$ either PDE2A2-GFP or, as a control, PDE2A1-GFP. Unlike PDE2A2-GFP, which shows a clear mitochondrial localization, PDE2A1-GFP distributes homogeneously in the cytosol (Supplementary Fig. 6). As shown in Fig. 4a, expression of PDE2A2-GFP, but not of PDE2A1-GFP or of the mitochondria-targeted mito-mEmerald (Supplementary Fig. 6) significantly increased Parkin recruitment to depolarized mitochondria, both in untreated cells and in cells treated with Forskolin. At higher concentrations of Forskolin the amount of cAMP generated appears to overcome the hydrolytic activity of PDE2A2. In HEK293T cells the effect of CCCP on Parkin recruitment was also blocked by elevation of cAMP induced by β-adrenergic receptor stimulation with ISO (Fig. 4b). In these cells expression of PDE2A2-GFP counteracted the inhibitory effect of ISO, an effect that was not observed in cells expressing PDE2A1-GFP or mito-mEmerald (Fig. 4c, d). Notably, expression of PDE2A2-GFP, but not expression of PDE2A1-GFP or GFP alone, was sufficient to increase by two-fold the mitochondrial recruitment of Parkin in MEF$^{wt}$ even in the absence of CCCP (Fig. 4e and Supplementary Fig. 7). The effect of PDE2A2 on Parkin recruitment relies on the enhanced local hydrolysis of cAMP at the mitochondria as Parkin recruitment was not enhanced by expression of a catalytically inactive point mutant of PDE2A2 (PDE2A2_DN-RFP) or N-term+GAF-B-RFP (Fig. 4f and Supplementary Fig. 8) a truncated mutant of PDE2A2 that localizes to mitochondria (Supplementary Fig. 3d–f) and interacts with MICOS (Fig. 2b) but lacks the catalytic domain.

Recruitment of Parkin to mitochondria is one of the best-characterized mechanisms leading to mitophagy. To more directly assess whether PDE2A2 activity at MICOS modulates mitophagy, we measured co-localization of mitochondria and LC3 (Microtubule-associated proteins 1A/1B light chain 3B), a

protein involved in autophagy and commonly used as a marker of autophagosomes[41]. For this purpose we co-transfected MEF$^{WT}$ with LC3-GFP, mRFP as a marker for mitochondria, and PDE2A2, and found significantly enhanced LC3/mRFP co-localization, and thus mitophagy, when compared with expression of PDE2A1, confirming the selective involvement of isoform 2 of PDE2A (Fig. 4g). Notably, PDE2A2 expression did not affect the total number of LC3 puncta (Fig. 4h), which is a measure of global autophagy[42], confirming a role of PDE2A2 selectively in the regulation of mitophagy.

In further support of its role in the regulation of mitophagy, we found that overexpression of PDE2A2 significantly increased the number of cells showing mitophagy puncta as detected by the mito-QC reporter[43] (Fig. 4i, j). This effect was not recapitulated by overexpression of either PDE2A1 or PE2A3, indicating a specific role of the 2A2 isoform (Fig. 4i, j).

**PDE2A inhibition blocks Parkin recruitment and mitophagy.** We next assessed whether pharmacological inhibition of endogenous PDE2A activity results in reduced Parkin recruitment and inhibition of mitophagy. As shown in Fig. 5a, b, treatment with BAY prevented Parkin translocation to the mitochondria induced by CCCP, even in the absence of cAMP raising agents. The combination of 1 μM BAY and 1 nM ISO showed additive effect, resulting in more prominent reduction in Parkin recruitment compared to the individual treatments (Fig. 5a, b). The effect of both ISO and BAY relies on PKA-mediated phosphorylation as it was completely abolished in cells expressing the selective PKA inhibitor peptide PKI[44] (Fig. 5c). The ability of PDE2A inhibition to reduce mitophagy was confirmed in experiments where MEF$^{WT}$ expressing mito-QC were treated with CCCP alone or in combination with BAY or, as a control, the PDE3 inhibitor cilostamide (Fig. 5d, e), a selective PDE3 inhibitor that generates cAMP at the mitochondria, albeit at a significantly lower level (Fig. 5f).

The effects of pharmacological inhibition were recapitulated by genetic ablation of PDE2A. We compared MEF$^{WT}$ and MEF$^{PDE2A-/-}$ and found that PDE2A KO results in significantly attenuated Parkin recruitment both in the absence and in the presence of CCCP (Fig. 5g and Supplementary Fig. 9). While 100 nM ISO significantly reduced Parkin recruitment in CCCP-treated MEF$^{WT}$, as observed in HEK cells (Fig. 4b), no effect of ISO was observed in MEF$^{PDE2A-/-}$ (Fig. 5h, j), indicating that the effect of adrenergic stimulation is entirely recapitulated by PDE2A ablation. Consistently, inhibition of PDE2A with BAY reduced Parkin recruitment in CCCP-treated MEF$^{WT}$ but showed no effect in MEF$^{PDE2A-/-}$ (Fig. 5i, j).

To confirm a role for PKA-dependent phosphorylation of MIC60 in the reduced recruitment of Parkin observed in MEF$^{PDE2A-/-}$, we used MIC60 phospho-deficient (S528A) and phospho-mimetic (S528D and S528E) mutants at the PKA phosphorylation site[19]. Expression of MIC60$^{WT}$ and MIC60$^{S528A}$ significantly increased

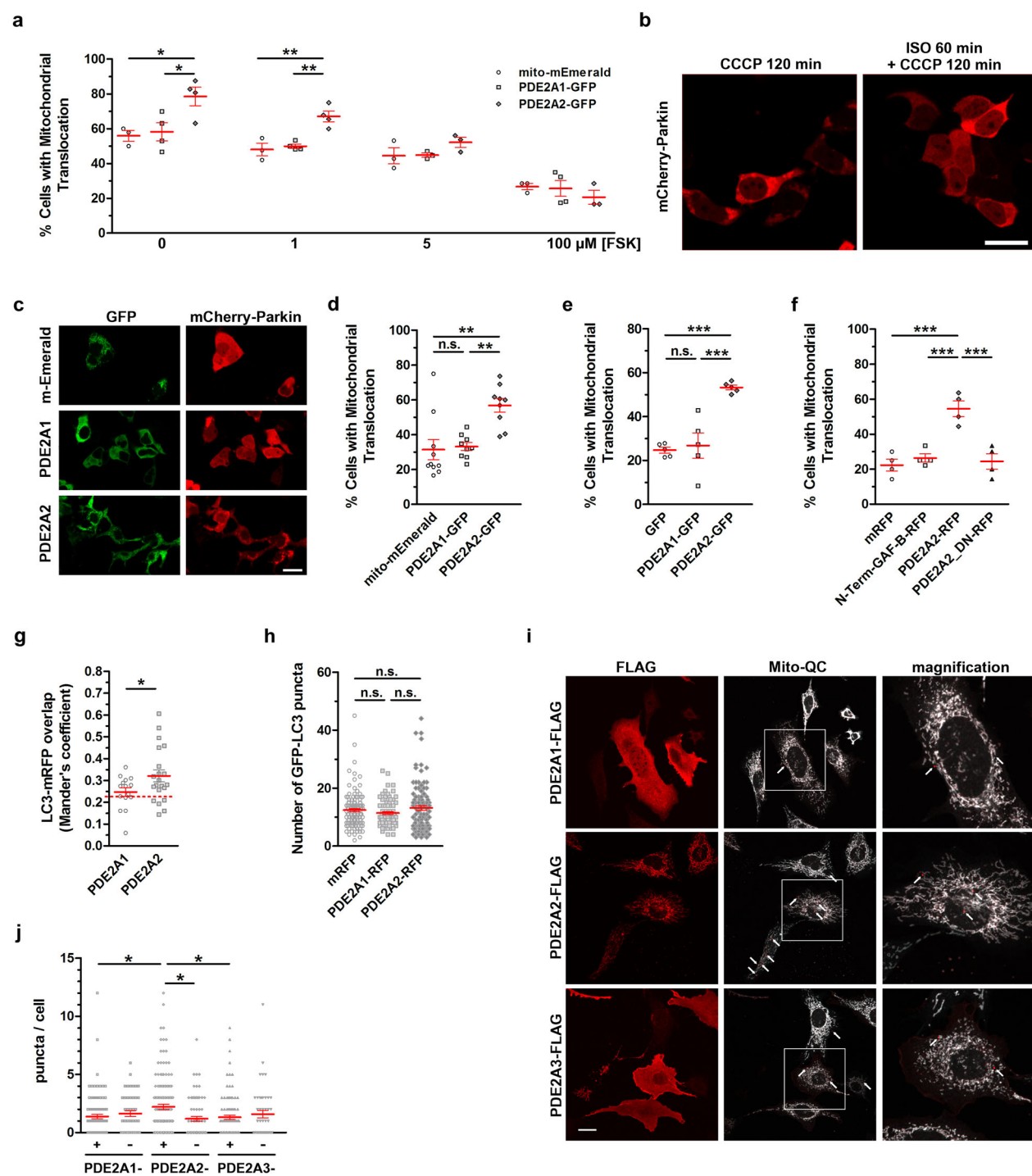

mCherry-Parkin recruitment in both MEF[WT] and MEF[PDE2A−/−] (Fig. 5k). In contrast, expression of MIC60[S528D] and MIC60[S528E] reduced the intracellular localization of mCherry-Parkin, consistent with the observation that PKA-mediated phosphorylation of MIC60 attenuates Parkin recruitment to the mitochondria[19]. In all cases, the difference in mCherry-Parkin recruitment between MEF[WT] and MEF[PDE2A−/−] was abrogated (Fig. 5k), confirming that PDE2A2 regulates Parkin recruitment via phosphorylation of MIC60 at the PKA site.

We also found that MEFs[PDE2A−/−], which have increased MIC60 phosphorylation (Fig. 3h, i), presented mitochondria with intact cristae morphology (Fig. 5l) supporting previous findings that PKA-dependent regulation of MICOS affects Parkin

recruitment independently from mitochondrial membrane structure[19].

## Inhibition of PDE2A modulates basal mitochondria clearance.
Treatment with CCCP results in severe mitochondria depolarization that is unlikely to reflect physiological conditions. To assess whether inhibition of PDE2A may impact mitochondria clearance in a physiological setting we treated MEF[WT] with BAY in otherwise untreated cells. Consistent with the higher level of PKA-dependent phosphorylation of MIC60 that we observed upon inhibition of PDE2 in untreated HEK293 cells (Fig. 3a, b and Supplementary Fig. 4) and Sh-SY5Y cells (Fig. 3d), inhibition

**Fig. 4 Effect of increased PDE2A2 activity on the recruitment of Parkin to the mitochondria and mitophagy. a** Number of MEF[WT] with mitochondrial Parkin. Cells expressing mCherry-Parkin and either mito-mEmerald, PDE2A1-GFP or PDE2A2-GFP were treated with Forskolin for 60 min followed by 180 min incubation with 10 μM CCCP. $n \geq 52$, $N = 3$-4. **b** HEK293T cells expressing mCherry-Parkin upon 120 min incubation with 10 μM CCCP or 60 min with 10 nM Isoproterenol (ISO) followed by 120 min incubation with 10 μM CCCP. **c** Representative images and summary data (**d**) showing number of HEK293T cells with mitochondrial Parkin. Cells expressing mCherry-Parkin and either mito-mEmerald, PDE2A1-GFP or PDE2A2-GFP were treated with 10 nM Isoproterenol (ISO) for 60 min followed by 120 min incubation with 10 μM CCCP. $n \geq 112$ cells, $N = 9$. **e** Number of MEF[WT] with mitochondrial Parkin, 24 h after co-transfection with mCherry-Parkin and either GFP, PDE2A1-GFP or PDE2A2-GFP. $n \geq 48$ cells, $N = 5$. **f** Number of MEF[WT] with mitochondrial Parkin, 24 h after co-transfection with GFP-Parkin and either mRFP, N-term-GAF-B-RFP, PDE2A2-RFP or PDE2A2_DN-RFP. $n \geq 89$ cells, $N = 4$. **g** Quantification of colocalisation of the signal generated by GFP-LC3 puncta and mRFP determined in MEF[WT] cells expressing either PDE2A1-FLAG or PDE2A2-FLAG. The broken line indicates level of colocalisation in cells not tranfected with PDE2A isoforms. $n \geq 15$ cells, $N = 3$. **h** Number of GFP-LC3 positive puncta in HeLa cells co-transfected with GFP-LC3 and either mRFP, PDE2A1-RFP or PDE2A2-RFP. $n \geq 79$ cells, $N = 3$. **i** Representative imeges and **j** summary data showing number of mitophagy puncta in MEF[wt] expressing mito-QC and transfected with PDE2A1-FLAG, PDE2A2-FLAG or PDE2A3-FLAG. The expression of PDE isoform was identified by immunostaining using an anti-FLAG antibody. As a control, puncta in cells from the same coverslip but negative for FLAG antibody signal are shown. Data are from $n \geq 46$ cells, $N = 3$. Scale-bars represent 20 μm for (**b**) and (**c**), 25 μm for (**i**). All values represent mean ± s.e.m. One-way ANOVA and Tukey's multiple comparison tests were performed for (**a**, **d**, **e**, **f** and **h**). Two-way ANOVA and Holm-Sidak's multiple comparison tests were performed for (**j**). Student's $t$ test was performed for (**g**). *$P < 0.05$; **$P < 0.01$; ***$P < 0.005$; ns = not significant.

of PDE2A was effective in reducing Parkin recruitment to mitochondria in MEF[WT] not treated with CCCP; by contrast, treatment with cilostamide did not show the same effect (Fig. 6a and Supplementary Fig. 10). Reduced overlap of LC3 puncta and mRFP (Fig. 6b) and unchanged overall number of LC3 puncta (Fig. 6c) confirmed that ablation of PDE2A activity specifically attenuates mitophagy in these cells. Notably, inhibition of PDE2A, but not of PDE3, also resulted in significant reduction of Parkin recruitment in inducible human pluripotent stem cell-derived dopaminergic neurons (Fig. 6d and Supplementary Fig. 11). The effect of PDE2A inhibition on basal mitophagy was further confirmed by the observation that MEF[WT] expressing mito-QC and treated with BAY, but not with cilostamide, show reduced number of mitophagy puncta compared to control cells (Fig. 6e, f).

**PDE2A inhibition controls PINK1 stability on mitochondria.** Parking localization to mitochondria is directly regulated by phosphatase and tensin homolog-induced kinase 1 PINK1[45] and CCCP treatment is known to stimulate autophosphorylation of PINK1 and its stabilization on depolarized mitochondria[46] (Fig. 7a, b). On the other hand, PKA-dependent phosphorylation of MIC60 destabilises MIC60-PINK1 interaction and results in a decline in the amount of PINK1 due to its rapid degradation by the proteasome[19], an effect that is recapitulated by treating HEK293T cells with 25 μM forskolin for 60 min (Fig. 7a, b). We therefore asked whether PDE2A inhibition regulates PINK1 stability on mitochondria through this mechanism. As shown in Fig. 7a, b, inhibition of PDE2A with BAY resulted in a small reduction in the amount of PINK1. The effect of PDE2A inhibition was however significantly amplified (60% reduction in total PINK1) when cells were simultaneously treated with 100 nM ISO and 1 μM BAY. In further support of the role of PDE2A2, we found that the effect of PDE2A inhibition on PINK1 stability was completely recapitulated by overexpression of the catalytically inactive mutant PDE2A2DN (Fig. 7c, f). PDE2A2DN contains 2 D to A point mutations at positions 685 and 796 in the enzyme catalytic site that completely abolish the ability to hydrolyse cAMP while maintaining the enzyme ability to participate in protein-protein interactions and anchor to subcellular sites[44,47]. Overexpression of this mutant is expected to displace endogenous active PDE2A2 at the mitochondria.

**PDE2A regulates adipocyte browning.** To assess the relevance of PDE2A2-dependent regulation of mitophagy in a physiologically relevant context, we focused on white adipocytes. Activation of

cAMP signaling in these cells, e.g., via norepinephrine (NE) stimulation, is known to lead to the transition from a fat storing (white) to an energy expending, mitochondria-rich (beige) phenotype, a process that involves attenuated mitophagy. We found that, as observed in HEK293 cells (Fig. 7a, b), CCCP treatment stabilizes PINK1 in white adipocytes (Fig. 8a, b) and inhibition of PDE2A potentiates the effect of ISO in promoting reduction of PINK1 levels (Fig. 8a, b). In addition, we found that BAY significantly enhanced the expression of nuclear thermogenic genes and mitochondrial markers in white adipocytes treated with NE, as shown both at the mRNA (Fig. 8c) and protein level (Fig. 8d, e). To further gauge the size of the mitochondrial pool, we also assessed expression of mitochondrially encoded genes relative to a nuclear housekeeping gene (*Hprt*)[48]. We found that inhibition of PDE2A enhances the effect of NE on overall mitochondrial gene expression, suggesting that PDE2A regulates mitochondrial content in adipocytes (Fig. 8f). Consistent with a role of PDE2A in the control of mitochondrial content via regulation of mitophagy, we found that inhibition of PDE2A significantly enhanced UCP1-dependent oxygen consumption in adipocyte treated with NE (Fig. 8g). In addition, we observed that inhibition of PDE2A in white adipocytes treated with NE further enhanced the decrease in lipid droplet size (Fig. 8h). All together, these data indicate that inhibition of PDE2A enhances cAMP induced 'beeing'[21–23] via promoting reduction of PINK1 levels.

**Discussion**
In this study we discovered that PDE2A2 interacts with MICOS at mitochondrial membranes; we established that, for this interaction to occur, PDE2A2 must be targeted to the mitochondria via its N-terminal domain (amino acids 1-201) and that the interaction itself involves the PDE2A2 regulatory GAF-B domain. We demonstrate that mitochondrial PDE2A2 controls a local pool of cAMP that regulates PKA-mediated phosphorylation of the MICOS component MIC60. We further demonstrate that the MICOS/PDE2A2/cAMP/PKA signalosome modulates Parkin recruitment to the mitochondria and mitophagy.

At least 100 different PDE isoforms are predicted to be expressed in cells[49]. The diversity of kinetics properties, substrate binding affinities, subcellular localization and regulatory mechanisms of individual PDE isoforms make this enzyme system ideally suited to fine-tune cAMP signals at a subcellular local level. PDEs are attractive therapeutic targets for precision medicine[8] as manipulation of the activity of individual PDE isoforms results in local cyclic nucleotide changes that are expected to impact specific cellular functions and limit side effects[8]. Dissecting the subcellular localization of individual PDE isoforms

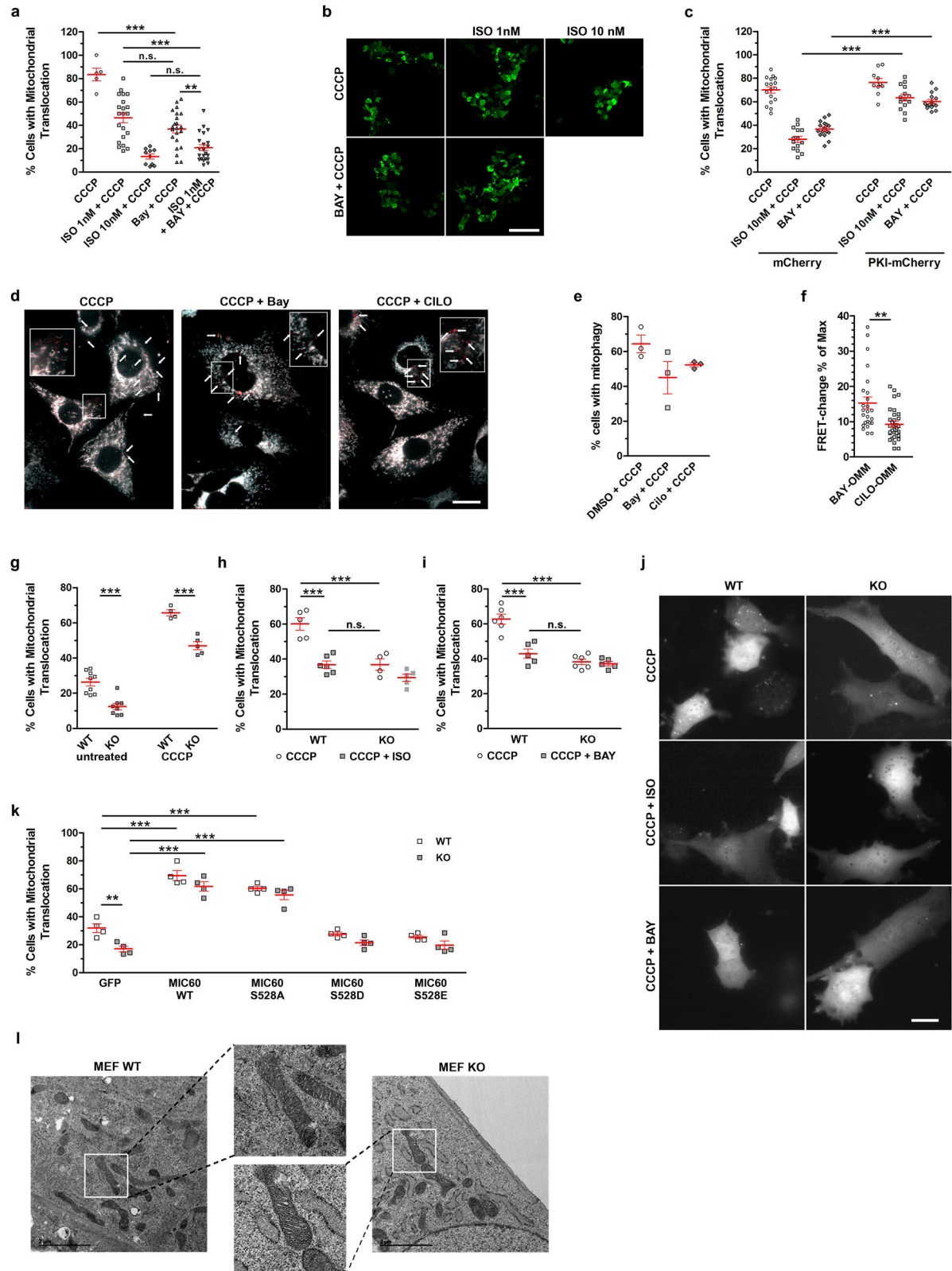

and defining their functional role at distinct sites is therefore important. One approach to this undertaking is to analyze the interactome of individual PDE isoforms.

We previously demonstrated that isoform 2 of PDE2A predominantly, but not exclusively, localizes to mitochondrial membranes where it controls Drp1 phosphorylation, mitochondria elongation and apoptosis[31]. Other studies have reported

localization of PDE2A2 to the mitochondrial matrix and a role in the regulation of oxidative phosphorylation[50,51]. However, the molecular basis for the specific sub-mitochondrial localization of PDE2A2 had not been explored. Our GO category enrichment analysis of the PDE2A2 interactome shows a significant enrichment of both plasma membrane and intracellular membranes categories (Supplementary Data 2), supporting the localization of

**Fig. 5 Effect of PDE2A2 inhibition on Parkin recruitment and mitophagy. a** Mitochondrial Parkin in HEK293T expressing mCherry-Parkin, treated with ISO or BAY-60 for 1 h followed by 2 h treatment with 10 μM CCCP. $n \geq 62$ cells, $N = 3$. **b** Representative images for (A). Scale bar: 100 μm. **c** Mitochondrial Parkin in HEK293T expressing GFP-Parkin and either mCherry or PKI-mCherry, upon 1 h treatment with ISO, BAY-60 or their combination followed by 10 μM CCCP treatment for 2 h. $n \geq 211$ cells, $N = 4$. **d** Representative images and (**e**) summary data showing MEF$^{WT}$ expressing mito-QC treated with CCCP in the absence (DMSO) or presence of BAY-60 or cilostamide. ($n \geq 112$ cells, $N = 3$. Scale bar: 20 μm. **f** cAMP changes in MEF$^{WT}$ expressing the mitochondrially targeted FRET reporter OMM-H90[31] treated with either 1 μM BAY-60 or 10 μM Cilostamide. FRET change is expressed as relative to maximal FRET change at saturation (25 μM Forskolin + 100 μM IBMX). **g** Mitochondrial Parkin in MEF$^{WT}$ or MEF$^{PDE2-/-}$ (KO) expressing mCherry-Parkin untreated (NT) or treated with 10 μM CCCP for 3 h. $n \geq 46$ cells, $N = 5$. **h** Mitochondrial Parkin in MEF$^{WT}$ or MEF$^{PDE2-/-}$ expressing mCherry-Parkin. Treatment: 10 μM CCCP for 3 h with or with 100 nM ISO for 1 h followed by 3 h treatment with 10 μM CCCP. $n \geq 107$ cells, $N = 3$. **i** Mitochondrial Parkin in MEF$^{WT}$ or MEF$^{PDE2-/-}$ expressing mCherry-Parkin and treated with 10 μM CCCP for 3 h or 1 μM BAY-60 for 1 h followed by 2 h treatment with 10 μM CCCP. $n \geq 165$ cells, $N = 3$. **j** Representative cells for (**g–h**). Scale bar: 20 μm. **k** Mitochondrial Parkin in MEF$^{WT}$ or MEF$^{PDE2-/-}$ expressing mCherry-Parkin and co-expressing either GFP, MIC60 wt, or the MIC60 mutants S528A, S528D or S528E. $n \geq 50$ cells, $N = 4$. Values are expressed as mean ± s.e.m. One-way ANOVA and Tukey's multiple comparison tests were performed for (**a**) and (**e**). Two-way ANOVA and Tukey's multiple comparison tests were performed for (**c**, **g**, **h** and **i**). Two-way ANOVA with Holm-Sidak's multiple comparison test for **k**. Student's $t$-test for **f** **$^{**}P < 0.01$; $^{***}P < 0.005$; ns = not significant. (**l**) Transmission electron micrographs of MEF$^{WT}$ (PDE2$^{+/+}$) and MEF$^{PDE2-/-}$ (PDE2$^{-/-}$) cells. Scale bar: 2 μm.

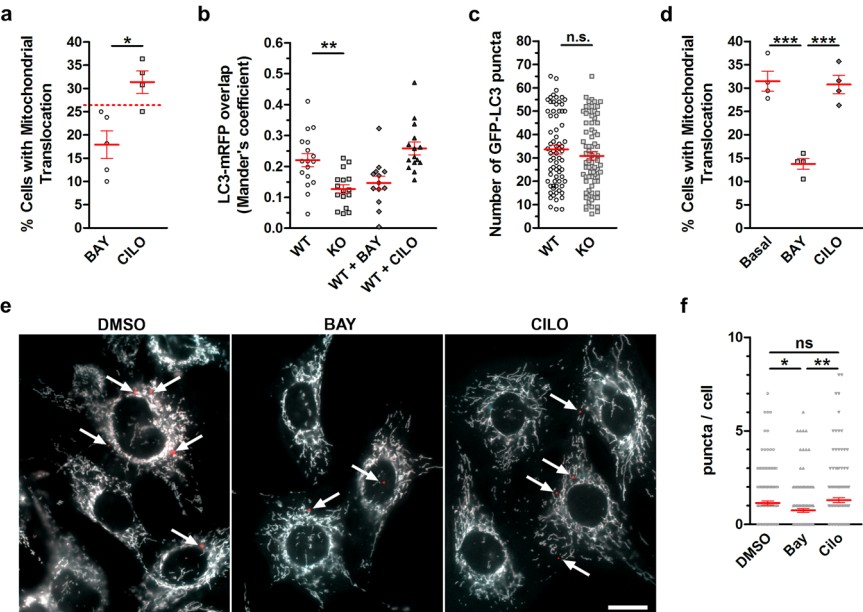

**Fig. 6 Effect of PDE2A2 inhibition on mitochondria clearance. a** Quantification of mitochondrial Parkin in MEF$^{WT}$ expressing mCherry-Parkin and treated with 1 μM BAY-60 or 10 μM Cilostamide (CILO) for 2 h. Dotted line indicates mitochondrial Parkin in untreated control. $n \geq 33$ cells, $N = 4$. **b** Overlap of GFP-LC3 puncta with mitochondria in MEF$^{WT}$ or MEF$^{PDE2-/-}$ expressing GFP-LC3 and mRFP under basal conditions or treated with 1 μM BAY-60 or 10 μM Cilostamide for 2 h. $n \geq 15$ cells, $N = 3$. **c** Number of GFP-LC3 puncta in MEF$^{WT}$ and MEF$^{PDE2A-/-}$ transfected with GFP-LC3 under basal conditions. $n \geq 71$, $N = 5$. **d** Quantification of mitochondrial Parkin localization in human iPSC-derived dopaminergic neurons expressing mCherry-Parkin under basal conditions or after treatment with 1 μM BAY-60 or 10 μM Cilostamide for 2 h. $n \geq 67$cells, $N = 4$. **e** Representative images and **f** quantification of mitophagy puncta in MEF$^{WT}$ cells expressing mito-QC and treated with 1 μM BAY or 1 μM cilostamide. $N \geq 157$cells, $N = 3$. Scale bar in (**e**) represents 20 μm. All values are expressed as mean ± s.e.m. One-way ANOVA and Tukey's multiple comparison tests were performed for (**b**, **d** and **f**). Student's $t$-test statistical analysis was performed for (**a** and **c**). $^{*}P < 0.05$; $^{**}P < 0.01$; $^{***}P < 0.005$; ns = not significant.

PDE2A2 at multiple subcellular sites. In the category enrichment analysis of our PDE2A2 pull-down, we found the category 'mitochondrial membranes' significantly enriched (Supplementary Data 2), while only one protein annotated to the GO category 'mitochondrial matrix' (BCL2L1, a protein involved in apoptosis and mostly abundant at the MOM[52]), which suggested that PDE2A2 more likely localizes at mitochondrial membranes.

Within the 'mitochondrial membranes'-annotated proteins, the MICOS components MIC60 and MIC19 were particularly interesting potential interactors of PDE2A2. MICOS is an evolutionarily conserved multiprotein system localized at the MIM and essential for cristae junction formation and maintenance[53]. In addition, MICOS establishes contact sites between the MIM and the MOM that facilitate mitochondrial protein import and are involved in phospholipid metabolism and distribution within

mitochondria[54]. Interestingly, our interactome analysis shows significant interaction of PDE2A2 also with the SAM complex, which is involved in mitochondrial protein import, as well as with proteins involved in the regulation of membrane lipid exchange and phospholipid transport (Supplementary Data 1 and Supplementary Data 2). Whether manipulation of PDE2A2 activity may impact on these functions of MICOS warrants further studies.

MIC60 is an integral MIM protein and a central component of MICOS[53]. Evidence supports its role as a scaffold that coordinates multiple mitochondrial functions at cristae junctions[52] and undergoes phosphorylation-dependent regulation. PINK1-mediated phosphorylation of MIC60 is required to maintain cristae architecture[55] whereas its phosphorylation by PKA inhibits recruitment of Parkin to damaged mitochondria[19]. Here, we demonstrate that genetic KO or pharmacological inhibition of

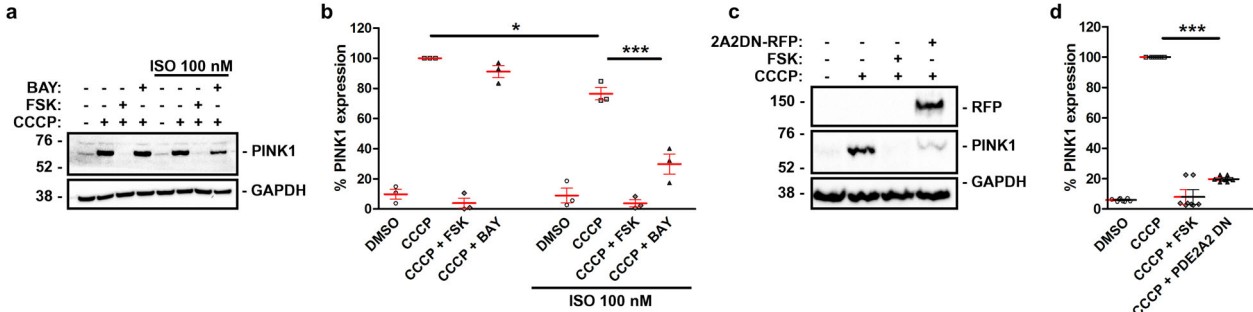

**Fig. 7 PINK1 stability is regulated by PDE2A2 activity. a** Representative western blot and (**b**) summary quantification of PINK1 levels in HEK293 whole-cell lysates treated with forskolin 25 µM, BAY-60 1 µM or ISO 100 nM, as indicated, followed by CCCP 10 µM treatment for 2 h, as indicated. $N = 3$. **c** Representative western blot and (**d**) summary quantification of PINK1 levels in HEK293 cells treated with CCCP 10 µM and forskolin 25 µM, as indicated, in the absence or presence of PDE2A2-DN-RFP. $N = 4$. All values represent mean ± s.e.m. and are expressed as relative to the level of PINK1 detected in cell treated with CCCP only. One-way (**d**) and two-way (**b**) ANOVA and Tukey's multiple comparison test were used *$P < 0.05$; **$P < 0.01$; ***$P < 0.005$.

PDE2A drives increased PKA-dependent phosphorylation of MIC60, reduces Parkin recruitment to the mitochondria and attenuates mitophagy, while increased activity of PDE2A2, but not of PDE2A1 or PDE2A3, shows the opposite result. This effect likely involves destabilization of MIC60 interaction with PINK1 at the MOM as a consequence of PKA phosphorylation of MIC60, and the consequent reduction of PINK1 levels by the lysosomes[19] as we find that suppressing the activity of PDE2A2 at the mitochondria results in reduction of PINK1 protein levels. In line with previous evidence that increased PKA phosphorylation of MIC60 does not impact on cristae remodeling[19], we show that genetic ablation of PDE2A2 decreases mitophagy without affecting mitochondrial structural integrity.

Inhibition of Parkin recruitment and mitophagy by the mitochondrial PDE2A2 but not the cytosolic isoform PDE2A1 or the plasma membrane-targeted PDE2A3 strongly indicates that PDE2A2 affects mitophagy via local modulation of cAMP at MICOS. Our data also demonstrate that the effect of ISO stimulation on Parkin recruitment is completely recapitulated by genetic ablation of PDE2A. This finding suggests that adrenergic stimulation controls mitophagy via the local change of cAMP levels at the mitochondria and reveals a central physiological role of PDE2A2 in the modulation of mitochondria clearance.

The regulation and roles of Parkin-dependent mitophagy are not completely understood. Several studies show that a reduction in the mitochondrial membrane potential leads to mitophagy through stabilization of PINK1 at the MOM and subsequent recruitment of Parkin[41,45,56]. In most in vitro studies, however, highly toxic mitochondria uncouplers have been used to induce Parkin recruitment and the nature of the triggers acting upstream of PINK1-Parkin recruitment in physiological conditions has been elusive. As a consequence, the relevance of this pathway in the regulation of mitophagy in vivo remains controversial[57] and recent evidence suggests that Parkin-dependent mitophagy may be context- and cell type-specific[58]. Equally uncertain are the exact function and the signaling triggers driving basal mitophagy, a process that occurs extensively across many tissues in the absence of any overt stress[59,60]. Here we show that, in a number of cell types, inhibition of PDE2A increases phosphorylation of MIC60, reduces Parkin recruitment and attenuates mitophagy even in the absence of any cAMP raising stimulus indicating that PDE2A2-dependent hydrolysis of cAMP at the MICOS complex regulates basal mitophagy. It is interesting to note that SKIP, an AKAP that selectively binds PKA type I holoenzymes[25], interacts with MIC19[61]. Although in our study we did not directly address the involvement of SKIP-anchored PKA in the phosphorylation of MIC60, it is tempting to speculate that PDE2A2 is part of a

MICOS/SKIP/PKA-I signalosome at the MIM. This possibility is particularly intriguing as SKIP can simultaneously bind two molecules of the PKA type-I holoenzyme[25], which is more sensitive to cAMP than the PKA type-II isoform. A signalosome organized by SKIP and MICOS at the MIM could therefore experience significant PKA activity even at relatively low cAMP concentrations, as one may expect in basal conditions. PDE2A2 catalytic activity is potently activated by cGMP. As cGMP levels vary in response to ATP changes[62] as well as physiological and pathological stressors[63], PDE2A2 provides a means to regulate mitochondria clearance in response to metabolic fluctuations and stress stimuli.

The cAMP-PKA pathway plays an outstanding role in adipocytes. Apart from lipolysis—the release of free fatty acids from adipocytes—cAMP signaling controls the metabolic phenotype and mitochondrial content of these cells[21,64]. Upon activation of cAMP signaling (e.g., via noradrenergic stimulation), white adipocytes change their phenotype from a fat storing to an energy expending cell, in a process known as "browning", and become so-called beige fat cells[22,23]. Beige adipocytes are characterized by expression of thermogenic genes (including UCP1), increased number of mitochondria, and attenuated Parkin recruitment and mitophagy. Altogether, these mechanisms regulate energy homeostasis and protect from diet-induced obesity[41,65]. Here we show that PDE2A plays an important role in this process. Interestingly, in vivo, $\beta_3$-adrenergic stimulation leads to impaired recruitment of Parkin, reduced mitophagy and retention of mitochondria-rich beige adipocytes and thermogenesis via cAMP and PKA activation[40]. This suggests that catecholamine-induced changes in cAMP levels can trigger Parkin recruitment to mitochondria in the absence of a mitotoxic stress. Our results show that, in several cell types, PDE2A2 can modulate Parkin recruitment to mitochondria in the absence of CCCP indicating that a local change of cAMP levels at MICOS and a change in the level of MIC60 phosphorylation may be sufficient to modulate Parkin translocation. We further show that inhibition of PDE2A enhances browning of white adipocytes induced in vitro by noradrenaline via destabilization of PINK1, supporting the notion that a local increase in cAMP in the absence of mitotoxic stress can reduce Parkin recruitment and limit mitophagy in these cells. Further studies will be required to confirm the involvement of MIC60 phosphorylation in adipocyte browning. However, our findings are of general relevance for adipose tissue, since several studies have documented mitochondrial loss in white adipose tissue of obese and diabetic mice[66,67]. Understanding regulation of autophagy and mitochondrial content in white and beige adipocytes has value for developing novel therapies for metabolic

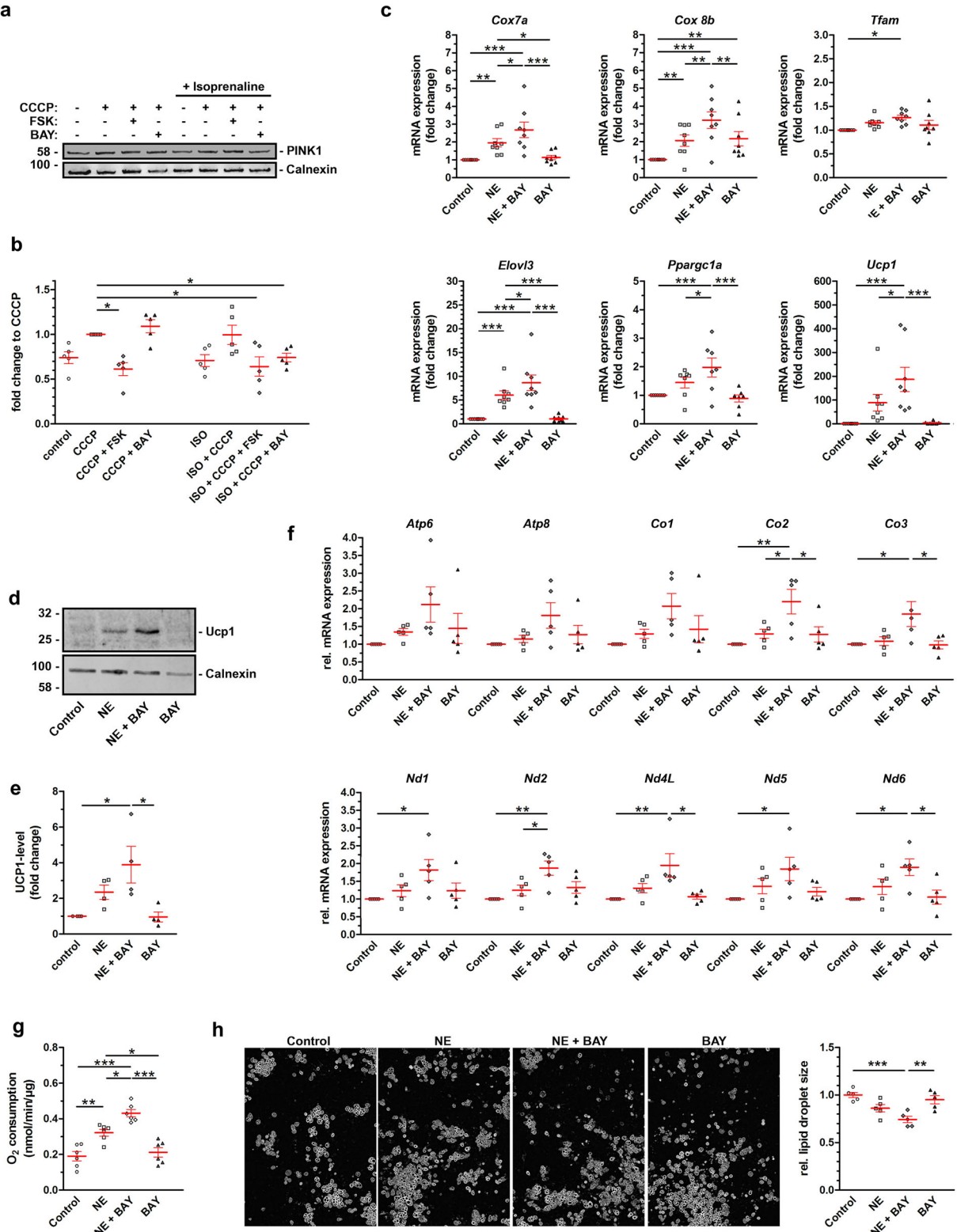

disorders. Our results suggest that PDE2A inhibition can promote a 'fat-burning' phenotype and may be an effective approach to retain thermogenic beige adipocytes to control obesity.

## Methods

**Reagents and antibodies**. Bay 60-7550 was from Cambridge Bioscience. Staurosporin was from Calbiochem. FuGeneHD was from Promega. Phosphate-Buffered Saline (PBS), Penicillin/Streptomycin, Trypsin 0.05%, Glutamine and

Lipofectamine 2000 were from Invitrogen. Unless otherwise noted, all other reagents were obtained from Sigma-Aldrich. Antibodies against the following proteins were used as recommended by the manufacturers: FLAG (Cell Signaling, 8146, 1:1000; Cell Signaling, 8146, 1:1000), GAPDH (SCBT, sc166574, 1:1000; Proteintech, 60004-1-Ig, 1:1000), GFP (SCBT, sc9996, 1:1000; Abcam, ab6556, 1:1000), MIC60 (Abcam, ab110329, 1:500), Parkin (Cell Signaling, 4211, 1:500), PDE2A (Proteintech, 55306-1-AP, 1:1000), phospho-(Ser/Thr) PKA substrate (Cell Signaling, 2964, 1:1000), PINK1 (Cell Signaling, 6946, 1:500), RFP (Chromotek, 6G6, 1:1000; Abcam, ab62341, 1:1000), SAMM50 (Atlas Antibodies, HPA042935, 1:1000) and α-Tubulin (Abcam, ab18251, 1:5000); horseradish-peroxidase-

**Fig. 8 Effect of PDE2A2 inhibition on adipocyte browning. a** Representative western blot and **b** quantification of PINK1 in primary white adipocytes treated with forskolin or BAY-60 in the presence of the mitochondrial uncoupler CCCP with or without stimulation of the beta-adrenergic receptor with Isoprenaline (ISO). The amount of PINK1 is normalized to Calnexin and is shown as fold change relative to levels in CCCP-treated cells. Data are mean ± s.e.m, $N = 5$. **c** mRNA expression of nuclear-encoded mitochondrial and thermogenic genes in primary white adipocytes treated with 1 μM NE, 1 μM BAY-60 or 1 μM NE + 1 μM BAY-60. mRNA level is shown as the fold change relative to levels in non-treated cells, $N = 7$. **d** Representative western blot and **e** quantification of UCP1 from primary white adipocytes treated with 1 μM NE, 1 μM BAY-60 or 1 μM NE + 1 μM BAY-60. The amount of UCP1 is shown as the fold change relative to levels in non-treated cells. $N = 4$. **f** mRNA expression of mitochondria-encoded genes in primary white adipocytes treated with 1 μM NE, 1 μM BAY-60 or 1 μM NE + 1 μM BAY-60. mRNA level is shown as the fold change relative to levels of *Hprt* $N = 5$. **g** UCP1-dependent oxygen consumption of primary white adipocytes treated with with 1 μM NE, 1 μM BAY-60 or 1 μM NE + 1 μM BAY-60, $N = 6$. **h** Representative images (left) and quantification (right) of the average lipid droplet size of primary white adipocytes treated with with 1 μM NE, 1 μM BAY-60 or 1 μM NE + 1 μM BAY-60, $N = 5$. All values are expressed as mean ± s.e.m. One-way ANOVA and Tukey's multiple comparison tests were performed for (**c**, **e**–**h**). Student's *t*-test statistical analysis was performed for (**b**). *$P < 0.05$; **$P < 0.01$; ***$P < 0.005$; ns = not significant.

conjugated goat anti-mouse or anti-rabbit secondary antibodies were also used (SCBT, sc2005, 1:5000; SCBT, sc2030, 1:3000), as well as antibodies conjugated with Alexa Fluor 647 (Thermo Fisher Scientific, A-21235, 1:250) and Alexa Fluor 532 (Thermo Fisher Scientific, A-11009, 1:250). Human pluripotent stem cell-derived dopaminergic neurons were from Cellular Dynamics (R1032).

**MS of the PDE2A2 interactome.** Constructs encoding for PDE2A2 or, as a control, EF-Ts (E.coli translation elongation factor Ts) tagged with a Strep-tag II epitope at the carboxyl terminal were generated via recombination cloning using the *StarGate*system (IBA BioTAGnology GmbH) following the manufacturer's instructions. For analysis of the PDE2A2 interactome HT-4 mouse neuroblastoma cells were transfected with either pESG-IBA103/PDE2 or pESG-IBA103/EF-Ts as a control (EF-Ts = *E. coli* translation elongation factor Ts). Lysates from transfected cells were used for immunoprecipitation with anti-StrepMab-Immo monoclonal antibody covalently coupled to Dynabeads ® Protein G. Elution fractions were separated on a 4–12% Bis-Tris Polyacrylamide gel for subsequent colloidal Coomassie staining, trypsinized and analyzed by MS (Linear Trap Quadrupole(LTQ)-Orbitrap Velos, Thermo Fisher Scientific) as previously described[68]. The MS data from two biological replicates were analyzed with the MaxQuant computational platform[69] (version 1.2.0.11). A false discovery rate of 1% was used for peptide and protein identification. The label-free quantification (LFQ) algorithm[33] was used for protein quantification; only unique peptides were taken into account for protein quantification. Re-quantify and Match between run were enabled for the analysis. MS data were further analyzed with the Perseus module of MaxQuant[70]. First, only identified by site, reverse and contaminant hits were removed from the interactome and only proteins pulled-down with PDE2A2 in both replicate experiments were considered for the analysis. Missing values in the control pull-down were replaced using the Imputation tool with default parameters. A two sample *t*-test was used to determine the interactome of PDE2A2, using a 10% FDR and S0 value of 0.2. For category enrichment analysis of the PDE2A2 interactome, a Fisher exact test with 1% FDR was used.

**Cell culture.** All experiments using primary cells comply with ethical regulations under the Animals (Scientific Procedures) Act 1986, section 5, and have been approved by the Home Office, UK.

HeLa, HEK293T and MEF cells were grown in DMEM + 10% FBS. Sh-Sy5y were grown in DMEM/F12 + 10% FBS. All cell lines were maintained in a humidified $CO_2$ (5%) incubator at 37 °C and were split every 2-3 days after reaching 80% confluence. Cells were tested each month for mycoplasma contamination by PCR. All plasmids were transfected into HeLa cells using FuGENE HD, into HEK293T cells using PEI or Lipofectamine 2000 and into MEFs or Sh-sy5y cells using jetPRIME, according to manufacturer's instructions.

**MEFs isolation.** Pregnant females obtained from crossing of heterozygous PDE2KO mice (line B6;129P2-Pde2atm1Dgen/H, provided by EMMA, UK), were sacrificed after 13 days of gestation (E.13 dpc) and the embryos collected, then placed into separate dishes containing phosphate buffered saline (PBS). A segment of the tail was removed for genotyping, whereas the head and the internal organs were removed. The remaining tissue was washed with PBS and incubated in trypsin at 37 °C for 5–10 min. The solution was pipetted until the tissue was completely homogenized and the isolated cells were plated onto T75 flasks. For transfection the cells were plated onto sterile 24 mm diameter coverslips and maintained at 37 °C for 24 h.

**Isolation and culture of primary murine white adipocytes.** Primary white adipocytes were isolated from 8–12 weeks old C57BL/6 mice[71]. Inguinal WAT (WATi) from three mice was dissected and digested in DMEM containing 0.5% BSA and collagenase type II at 37 °C and then centrifuged at 1000 rpm for 10 min. The resulting pellet was re-suspended and filtered using a 100 μm nylon mesh. The filtered solution was seeded into a T175 culture flask in DMEM supplemented with 10% FBS and 1% P/S (WA Growth medium) and kept at 37 °C and 5% $CO_2$. 24 h

after seeding, cells were washed with PBS and maintained in WA Growth medium at 37 °C, 5% $CO_2$. Medium was refreshed every day until cells reached confluency and subsequently cryopreserved.

Cells were plated at a density of 140,000 cells per well in a 6-well plate in growth medium. Cells were grown to confluency in growth medium, changing the medium every other day. Once confluent (Day -1), cells were maintained in WA Growth medium for 24 h. On day 0 differentiation of pre-adipocytes was induced for 48 h (Day 0–Day +2) by changing the medium to WA Induction medium (DMEM containing 5% FBS, 1% P/S, 1 nM T3, 0.172 μM insulin, 50 mg/ml L-ascorbate, 1 mM D-biotin, 17 mM panthothenate, 1 μM Rosiglitazone, 0.25 μM dexamethasone and 0.5 mM 3-isobutyl-1-methylxantine). From day +2 until day +12, cells were maintained in WA Maintenance medium (DMEM containing 5% FBS, 1% P/S, 1 nM T3, 0.172 μM insulin, 50 mg/ml L-ascorbate, 1 mM D-biotin, 17 mM panthothenate), refreshing it every other day. Starting on Day 0, cells were chronically treated with 1 μM NE, 1 μM BAY-60 or their combination. Medium and treatment were refreshed every other day. All cells were analyzed on Day +12, with the exception of PINK1 analysis that was performed on 80% confluent cells.

**Immunoprecipitation and Western blotting.** Cells were washed twice with PBS and lysed for 5 min with RIPA buffer containing cOmplete™ EDTA-free protease inhibitor cocktail (Roche) and PhosSTOP phosphatase inhibitor cocktail (Roche). Pellets were clarified by centrifugation at 10,000 rpm for 10 min at 4 °C and the supernatant and incubated at 4 °C for 2 h with 25 μL of anti-RFP/GFP/Myc antibody-linked beads (Chromotek) or anti-FLAG M2 beads (Sigma), respectively, or untagged beads (Chromotek), as a control. (For immunoprecipitation of endogenous proteins, the supernatant was pre-cleared by incubation with Protein-G agarose beads (Invitrogen). The supernatant was then incubated for 2 h with the appropriate antibody at 4 °C and overnight incubation with Protein-G agarose beads, as previously described[72]. Beads were pelleted by centrifugation for 1 min at $5000 \times g$, washed 3–5× with wash buffer (Tris-HCl 0.1 M, NaCl 0.3 M, Triton X-100 1% (v/v), pH 7.5) followed by elution with 1× sample buffer and spin-down. Whenever appropriate, a 5 min incubation at 90 °C was performed during elution. The immunoprecipitate was loaded and separated on SDS-PAGE gels and subsequently transferred to PVDF membranes (Millipore). After transfer, PVDF membranes were blocked for 1 h at room temperature in 5% skimmed milk (Sigma) TBS-T (Alfa Aesar)and incubated overnight at 4 °C with primary antibody diluted in 5% skimmed milk/TBS-T. After three washes with TBS-T membranes were incubated at room temperature for 1 h with a peroxidase-conjugated secondary antibody. After three washes with TBS-T, chemiluminescent detection was performed with a Compact X4 Automatic Processor (Xograph) and the signal was developed using ECL Western Blotting Substrate (Thermo Scientific). Quantification of band intensity was accomplished by densitometry using ImageJ software (http://rsbweb.nih.gov/ij/index.html). The extent of protein phosphorylation was calculated as the ratio between the density of the band corresponding to the phosphorylated protein and the density of the band corresponding to total protein. Values were normalised using GraphPad Prism software (GraphPad Software, Inc., CA, USA),

White adipocytes were homogenized in RIPA buffer containing the above mentioned protease and phosphatase inhibitors. After centrifugation ($10,000 \times g$, 30 min, 4 °C), the supernatants were collected, and protein concentration was determined with Bradford reagent. Western blot analysis was performed as described by the manufacturer with rabbit anti-uncoupling protein 1 (UCP-1, Thermo Fisher Scientific) and rabbit anti-calnexin antibodies (Anti-Calnexin, C-Terminal, EMD Milipore Corp, USA). The bands were visualized with an Odyssey imaging system (LI-COR Bioscience, Lincoln, NE, USA) with fluorescence-labeled secondary antibodies (Anti-rabbit IgG (H + L) – Dylight 800, 4× PEG Conjugate, Cell Signaling Technology), according to the manufacturer's protocol. UCP1 and PINK1 signals were normalized to Calnexin.

Uncropped images of all full blots are shown in the Supplementary Material.

**Live cell confocal imaging.** Fluorescence imaging was performed 24 h after cell transfection. Cells were kept at RT in Dulbecco's PBS (DPBS; Life Technologies)

and imaged on a Fluoview FV1000 microscope, which is an inverted IX81 confocal system (Olympus, Tokyo, Japan), using a 60X, NA 1.35 oil immersion UPlanSApo objective (Olympus). The microscope was equipped with a Becker and Hickl FLIM system and with fluorescence filters suitable for UV and green or red dyes. Images were acquired using FluoView viewer software (FV10-ASW software, Olympus) and processed using ImageJ.

**Parkin recruitment**. mCherry-Parkin-expressing MEFs were treated with the indicated reagents, followed by CCCP incubation (10 μM, 3 h) except where indicated. Intracellular localization of mCherry-Parkin was analyzed under fluorescence microscopy using MetaFluor software.

The extent of overlap between LC3-GFP and mRFP (mitochondrial marker) was quantified using Mander's coefficient (ImageJ, Coloc 2 plugin). LC3 puncta analysis was performed by auto-thresholding images (ImageJ) and quantifying the number of puncta.

**Immunofluorescence**. Cells were fixed with 8% paraformaldehyde in TBS (Tris-buffered saline) for 10 min, washed in TBS, permeabilized with 0.5% (v/v) Triton X-100 (TX) in TBS for 5 min and washed in TBS. Non-specific labeling was blocked with a 10 min incubation of 1% normal goat serum (NGS; v/v) in Antibody Diluting Solution (AbDil; TBS 0.1% TX, 2% BSA and 0.1% Sodiumazide) and all antibodies were diluted in AbDil 1% NGS. Primary antibodies were incubated for 60 min. After $3 \times 5$ min washes cells were treated with fluorescent-conjugated species-specific immunoglobulins (1:250 in AbDil 1% NGS) for 60 min. After $3 \times 5$ min TBS washes, coverslips were mounted in Ibidi mounting media and sealed with nail varnish.

**FRET imaging**. Cells expressing OMM-H90[31] were imaged on an inverted microscope (Olympus IX71) using a PlanApoN, 60X, NA 1.42 oil immersion objective, 0.17/FN 26.5 (Olympus, UK). The microscope was equipped with a CoolSNAP HQ$^2$ cooled CCD camera (Photometrics) and a DV$^2$ beam-splitter (MAG Biosystems, Photometrics) for simultaneous imaging of CFP and YFP emissions. The FRET filters used were: CFP excitation filter ET436/20×, dichroic mirror 455DCLP (all from Chroma Technology) in the microscope filter cube; dichroic mirror 505dcxr, YFP emission filter 535/30 m, CFP emission filter 480/30 m (Chroma Technology) in the beam splitter. Images were acquired using Metafluor software. Changes in cAMP concentration were monitored by measuring CFP (480 nm)/YFP (535 nm) fluorescence emission values upon excitation of the transfected cells at 430 nm. FRET changes are expressed as relative to maximal FRET change at saturation (25 μM forskolin + 100 μM IBMX).

**Mito-QC imaging**. MEF$^{WT}$ expressing mito-QC[43] were a generous gift from Dr Ian Ganley (University of Dundee, UK). Cells were seeded on 24 mm diameter coverslips, grown for 24 h in 6-well plates in DMEM, supplemented with 10% FBS (Gibco, Art. No. 10270), 100 units/mL penicillin, 100 μg/mL streptomycin, and 2 mM glutamine at 37 °C, 5% CO$_2$ and then subjected to the relevant treatments. Cells were washed three times, fixed with 4% Paraformaldehyd (15 min), washed again 3 times and then stored at 4 °C until imaged. Widefield image acquisition was performed with MetaFluor (Universal Imaging) on an Olympus IX 71 inverted microscope at 60x magnification (Plan Apochromat N 60X NA 1.42 oil immersion) equipped with a Photometrics HQ$^2$ camera. Excitation/emission wavelength was $500 \pm 10$ and $535 \pm 15$ for GFP and $562 \pm 20$ and $624 \pm 20$ for RFP. For immuno-fluorescence of cells expressing mito-QC and Flag-PDEs, cells were incubated with Anti-FLAG antibody (DYKDDDDK-Tag, Anti-Rabbit, Cell Signaling, Cat. No. 2368S, 02-2017, lot 12) 1:1000 in 1% BSA overnight at 4 °C or for 2 h at RT. Cells were then washed 3 times and incubated for 2 h (RT) with 1:400 secondary antibody (AlexaFluor 633, Goat-Anti-Rabbit, Cat.No. A21070, Invitrogen) in 1% BSA. Finally cells were washed again three times and stored light protected at 4 °C until imaged. All dilutions were in PBS containing 20 mM HEPES/pH 7.0. For quantitation, red-alone puncta were counted in at least ten fields of view from at least three independent experiments for each condition. A threshold of 3 or more red-alone puncta per cell was applied to the data to determine the number of cells undergoing mitophagy. Data were expressed as percent of cells with mitophagy or as average number of red-alone puncta per cell.

**Molecular biology**. Isolation of RNA from cells was performed by using innu-SOLV RNA Reagent. Final concentration of RNA was quantified using a Nanodrop Spectrophotometer. Following the manufacturer's instructions, 1000 ng of RNA were transcribed using the "Transcriptor First Strand cDNA Synthesis Kit" (Roche). mRNA of the target genes was amplified and quantified by using the transcribed cDNA. mRNA expression was assessed by qRT-PCR using a HT7900 instrument from Applied Biosystems and SYBR-Green PCR master mix (Applied Biosystems). Quantification of mRNA levels was performed based on the crossing point values of the amplification curves using the second derivative maximum method. *Hprt* (Hypoxanthine-guaninephosphoribosyltransferase) was used as an internal control.

The primer sequences used to amplify the target genes are shown in Supplementary Table 1

**UCP1-dependent respiration**. Primary white adipocytes were treated as described above (1 μM BAY, 1 μM NE). On day 12 the measurements were conducted using an Oxygraph 2 K (Oroboros Instruments). Samples were transferred to the oxygraph chamber containing 2 ml incubation medium (0.5 mM EGTA, 3 mM MgCl$_2$ 6H$_2$O, 60 mM K-lactobionate, 20 mM taurine, 10 mM KH$_2$PO$_4$, 20 mM HEPES, 110 mM sucrose and 1 g/l BSA, pH 7.1). In vitro respiration levels were recorded when reaching a steady state followed by addition of substrates (State 1: endogenous; state 2: Pyruvate-Malate-Glutamate [PMG]; state 3: Succinate; state 4: GDP;). Respiration rates were normalised to total protein content and UCP1 respiration was calculated from the difference between GDP and PMG respiration rates.

**Determination of lipid droplet size**. Primary white adipocytes were treated as indicated above. On day 12, images were acquired through automated digital microscopy by Cytation$^{TM}$ 5 (BioTek). Three images per well were automatically taken. Image processing and calculation of the average lipid droplet size was done using the Gen5$^{TM}$ software provided (min. object size: 1 μM; max. object size: 40 μM).

**Statistics and reproducibility**. Data are presented as means ± standard error of the mean. The significance of differences between multiple groups were compared by one-way or two-way analysis of variance (ANOVA), as appropriate, followed by a Tukey's or Sidak's multiple comparisons tests. Two-group analysis was performed by two-tailed Student's *t*-test. Number of replicates is indicated in the figure legends: *N* indicates number of biological replicates; n indicates number of observations. Differences were considered significant at $p < 0.05$, with *$p < 0.05$; **$p < 0.01$; ***$p < 0.005$; ns = not significant. All data were analysed using GraphPad Prism 5.01 or Prism 8.43 software

**Reporting summary**. Further information on research design is available in the Nature Research Reporting Summary linked to this article.

## Data availability

All data generated or analysed during this study are included in this published article (and its supplementary information files). Source data behind the graphs can be found in Supplementary Data 3. The mass spectrometry proteomics data have been deposited to the ProteomeXchange Consortium via the PRIDE[73] partner repository with the dataset identifier PXD020486.

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

## Acknowledgements

The authors would like to thank Dr Ian Ganley for providing the mito-QC MEFs and Prof Miles Housley for critically reviewing an early version of the manuscript. This study was supported by the British Heart Foundation (PG/10/75/28537, PG/15/5/31110, RG/17/6/32944 and RG/12/3/29423), the BHF Centre of Research Excellence, Oxford (RE/08/004 and RE/13/1/30181) to M.Z., Medical Research Council (MR/K501256/1) and St. John's College to M.J.L. and Cancer Research UK grant A12935 to S.Z.

## Author contributions

M.J.L., L.R-S., F.G., A.F., A.K., Y-C.C., G.S., S.Z., and M.Z. designed experiments and analysed data. M.J.L., L.R-S., F.G., A.K., Y-C.C., G.S., S.P., N.L., and E.P. performed experiments and analysed data. M.Z. and M.J.L. wrote the manuscript. M.Z. conceived research question and oversaw the entirety of research.

## Competing interests

The authors declare no competing interests.

## Additional information

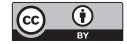

