## [Peer Review File · Communications Biology]

Reviewers' comments:

Reviewer #1 (Remarks to the Author):

Comments to the authors

In this manuscript, the authors found a mitochondrial cAMP-degrading enzyme PDE2A2 as a positive regulator of the Parkin recruitment. Considering about the PKA-mediated inhibition of the Parkin recruitment through the Mic60 phosphorylation (Akabane et al., Mol. Cell, 2016), the authors argument is consistent with previous findings. However, the authors should present the representative image data about the Parkin recruitment after the PDE2A2 modulation at least for their key findings, not only by quantified bar graphs (Fig. 4-6). Additionally, following points need to be addressed.

Major points

1. In Fig. 1B, the mitochondrial localization of PDE2A2 is convincing. However, it seems there are conflicting observations about the PDE2A2 localization within mitochondria (line 73-76, on page 2). Thus, it is recommended to examine the PDE2A2 localization within mitochondria using Proteinase K protection assay of isolated mitochondria.
2. In Fig. 2B, it is better to show that N-terminal + GAF-B-RFP mutant localizes in mitochondria. It seems it is missing in Supplemental Fig. S2.
3. The image in Fig. 4A was already showed in the previous paper. More importantly, some of the Parkin recruitment data after the PDE2A2 modulation in Fig. 4-6 should be presented by the representative image data, not only by bar graphs.
4. It has been shown that PKA-mediated Mic60 phosphorylation inhibits the PINK1 accumulation. Thus, it is better to show the PINK1 blot after the PDE2A2 modulation.

Minor points

1. In Fig. 3A, it is better to mention that the effect of BAY only can be observed in the lower panel (long exposure) of phospho-Mic60 blot.
2. In Fig. 3D and H, why the phospho-Mic60 blot shows ladder bands even in IP blot? At least, it is better to insert the molecular weight on this blot to show that the arrow head precisely indicates the immunoprecipitated Mic60.

Reviewer #2 (Remarks to the Author):

In their manuscript entitled "Phosphodiesterase 2A2 regulates mitochondria clearance via Parkin-dependent mitophagy", Lobo et al. report and characterize an interaction between the cAMP phosphodiesterase PDE2A2 and the MICOS subunit MIC60, a known PKA substrate. Accordingly, the authors go on to demonstrate a supportive role for PDE2A2 in parkin-dependent mitophagy using well-characterized assays that are typical for that field. The authors end on data demonstrating a role for PDE2A2 in the process of fat browning, which is linked to mitochondrial function.

The experimentation is generally robust, and the mass spectrometry analysis certainly serves as a resource to study the PDE2A2 interactome. I appreciated the authors' dissection of the MIC60-

PDE2A2 interaction. While they further develop this relationship in the context of parkin-dependent mitophagy, it was unclear to me if much was gained mechanistically on top of what was already known regarding the role of PKA in the latter process. Additionally, the findings concerning the role of PDE2A2 in fat browning at the end of the manuscript seemed out of place and almost counter-intuitive. I have outlined specific points below:

Major points

1. The link between the MIC60/PDE2A2/PKA axis and mitophagy followed closely with work done by Akabane et al. published several years ago (PMID 27153535) showing that PKA phosphorylates MIC60 and inhibits mitophagy. While it was difficult for me to see what was gained mechanistically by further implicating PDE2A2 in this process, the authors seemingly ignored completely the main finding of that paper: that phosphorylation of MIC60 by PKA prevents PINK1 accumulation on mitochondria. The authors should expect to see a similar modulation of PINK1 via genetic/pharmacological inhibition of PDE2A2. Is this so?
2. It was difficult for me to appreciate the data concerning fat browning (Fig. 6F and G). These data immediately follow studies on mitophagy (i.e. mitochondrial degradation). It is my understanding that the authors are suggesting that PDE2A2 facilitates mitochondrial turnover via autophagy, and this may explain the increased mitochondrial load of brown fat vs. white adipocytes upon BAY-60 treatment. The authors are however measuring mRNA, implying that mitochondrial biogenesis is in fact increased. Additionally, does PDE2A2 inhibition also increase mitochondrial biogenesis in the cell culture models in which mitophagy is assessed?
3. Dissection of the MIC60-PDE2A2 relationship – arguably the crux of the manuscript – was fairly straightforward and logically-approached. The authors ultimately suggest via truncation experiments that the GAFB domain targeted to the IMS is sufficient to bind MICOS. Determining whether or not a chimera of GAFB and an IMS-targeting sequence of another protein entirely (and not the PDE2A2 N-terminus) would demonstrably show that this domain is sufficient to bind MIC60.
4. Does overexpressing PDE2A2 suppress MIC60 phosphorylation (i.e. in relation to Fig. 3)?
5. The authors use the term “energized mitochondria” several times throughout the text to mean experiments performed in the absence of uncoupler. It is very likely that the specific organelles in question are depolarized, however, as a small number of mitochondria may become depolarized in a cell at a given time. This should be made clearer in the text. If the authors truly believe that these parkin-positive mitochondria have a membrane potential, they should demonstrate this by using a dye such as TMRM.

Minor points

1. It would benefit the reader by initially explaining the function of each GAF domain.
2. MIC60 levels are not visible in the input for the IPs in Fig. 3. Can the authors provide longer exposures?
3. The authors should explicitly state that the antibody used in Fig. 3 is against PKA phosphorylation.
4. Can the authors provide information in Materials and Methods for how blots were quantified? This is particularly important in the case of IPs.
5. The authors should show accompanying microscopy images for the data quantified in Fig. 5A-F and Fig. 6A-E. Moreover, are the cells in Fig. 5F treated with CCCP? If not, is it not surprising that

>60% of cells overexpressing MIC60 show parkin translocation? Is this translocation only partial?

Reviewer #3 (Remarks to the Author):

PDE2A, a cyclic nucleotide phosphodiesterase with dual specificity to cAMP and cGMP, exhibits three cellular isoforms that differ in their N-terminal sequences and their subcellular localizations. Significantly, PDE2A protein family has been implicated in regulating various cellular functions including cellular respiration, inflammatory reactions and membrane-related functions. Notably, PDE2A2 is predominantly a mitochondrial protein that has been implicated in regulating Drp1 phosphorylation, mitochondrial-mediated apoptosis and oxidative phosphorylation. In this work, the authors find an interaction between PDE2A2 and MICOS factors MIC60 and MIC19. MIC60 is an integral IMM membrane, which is a substrate for PKA and the PKA-dependent phosphorylation of PDE2A2 attenuates Parkin recruitment to the mitochondria and inhibits subsequent mitophagy. Ablation of PDE2A2-MIC60 interaction, genetically via the use of PDE2A knockouts, appears to increase the PKA-dependent phosphorylation of MIC60 and hence mitigate parkin recruitment and inhibit mitophagy. The authors rationalize that PDE2A-mediated inhibition of parkin recruitment might proceed through local modulation of cAMP levels at MICOS. The authors also show that PDE2A inhibition can promote a fat-burning phenotype through transition from white to beige adipocytes phenotype, although this part is somewhat superficial and disconnected from the rest of the story. Overall, while this study represents a considerable amount of solid work with experiments performed in several cellular models including immortalized cell lines and primary cells, there are still several points that needs to be addressed.

1- The authors claim that they identified the interactome of PDE2A2 specifically, and not other members of PDE2A. However, it is not clear to this reviewer why the authors used Elongation factor thermo stable protein (EF-T) from *E. coli* as negative control for the strep-tag pull down, rather than using the sequence encoding other members of PDE2A, the negative control of the pull down should have been sequence encoding PDE2A member other than PDE2A2. As the authors already indicated in the context of their co-IP's experiments, some mitochondrial substrates are strictly PDE2A-isoform dependent and hence distinguishing the specific substrates for each isoform might be feasible.

2- I'm a little confused by the mass spec results. In Fig. 1A, MIC60, MIC19 and SAMM50 appear prominent. However, when I scanned the supplemental data 1, I could not even find MIC60 or MIC19 among the list. SAMM50 was not even among the top 50 hits. While it makes sense to focus on the MICOS complex biologically, it makes one wonder about the rationale or usefulness of the mass spec data in the logic of this story.

3- I have the same comment for supplemental data 2. It seems a bit construed to say that mitochondrial membrane was selectively enriched in the GO analysis when it was ranked 21th out of 26 pathways.

4- P15, Lines380-381, "overall the above data support a role of PDE2A2 in regulation of" this statement is not strictly true, because the authors in this part investigated the role of PDE2A2 on regulation of PKA-mediated phosphorylation of MIC60 using BAY60-7550, an inhibitor for PDE2 members and not sole inhibitor for PDE2A2. Further, the genetic results obtained from PDE2A knockout that is devoid of the all the three isoforms of PDE2A (PDE2A1, PDEA2 and PDEA3). Thus, given this context of results, it is hard to dissect the distinct and strict role of PDE2A2 in regulation of PKA-phosphorylation of MIC60 without taking in consideration the role of other PDE2A's. In vitro kinase assays and/or overexpression of different forms PDE2A2 (active and kinase dead) might help in solidifying this part. At the very least, this point should be discussed.

5- While the authors indicated that PDE2A inhibition can increase the PKA-dependent Phosphorylation of MIC60, reduce parkin recruitment to the mitochondria and attenuate

mitophagy, a number of molecular aspects are not clear, for instance, how local cAMP modulation, via modulating levels of PDE2A and MIC60 phosphorylation, signals to alter parkin recruitment and mitophagy, this should be at least pointed out clearly in the discussion section.

6- While the authors revealed the role of inhibition of PDE2A on PKA-mediated phosphorylation of MIC60, parkin recruitment and mitophagy, the role of PINK1 kinase, a crucial factor in PARKIN-dependent mitophagy has not been addressed. PINK1 has been shown to phosphorylate MIC60 (Please check: PMID: 29456190) and this paper should be discussed. Whether PINK1 plays a role in the reported PDE2A-mediated regulation of mitophagy is not yet clear. It may be informative to investigate or at least discuss how the modulation of PDE2A could affect PINK1 and vice versa.

7- Finally, the experiments on adipocyte browning are very superficial and premature. There is no clear link to MIC60 phosphorylation or to the PINK1/Parkin pathway to justify its inclusion in the abstract and discussion. Unless, these connections can be made, this part should be de-emphasized or ideally removed.

8- Overlap of mitochondrial markers with LC3 is not a great way to quantify mitophagy. Have the authors considered using more quantitative approaches such as mito-Keima or mitoQC reporter systems? I don't think it is essential for this paper but I would highly recommend it if the authors want to pursue this type of work.

A point-by-point rebuttal to the issues raised by the reviewers is included below.

Reviewer #1:

Major points

1. In Fig. 1B, the mitochondrial localization of PDE2A2 is convincing. However, it seems there are conflicting observations about the PDE2A2 localization within mitochondria (line 73-76, on page 2). Thus, it is recommended to examine the PDE2A2 localization within mitochondria using Proteinase K protection assay of isolated mitochondria.

Previous studies have indeed reported the localisation of PDE2A2 in different submitochondrial compartments. We have already performed, and published, proteinase K protection assays of isolated mitochondria showing localization of PDE2A2 to mitochondria membranes and predominantly to the MIM (Monterisi, S. *et al. eLife* **6**, 2017). Another group had previously published data showing localization of PDE2A2 to the mitochondrial matrix (Acin-Perez, R. *et al. J. Biol. Chem.* **286**, 30423–30432 (2011). Although in our hands a PDE2A2 signal in the matrix was not detectable, the presence of a fraction of the enzyme in this compartment cannot be excluded. We have now reworded this paragraph in the revised manuscript to more clearly summarise the previously published evidence.

2. In Fig. 2B, it is better to show that N-terminal + GAF-B-RFP mutant localizes in mitochondria. It seems it is missing in Supplemental Fig. S2.

In Fig S3F we now show by western blot using purified mitochondrial fractions that the N-terminal + GAF-B-RFP mutant localises to the mitochondria

3. The image in Fig. 4A was already showed in the previous paper. More importantly, some of the Parkin recruitment data after the PDE2A2 modulation in Fig. 4-6 should be presented by the representative image data, not only by bar graphs.

Panel A in Fig 4 has now been removed and in the revised manuscript we show representative images of cells for the majority of the data summarised in Fig 4-6. These new images are shown in panel 4C (and Figs S7, S8), panels 5B, 5L (and Fig S9) and Figs S10 and S11.

4. It has been shown that PKA-mediated Mic60 phosphorylation inhibits the PINK1 accumulation. Thus, it is better to show the PINK1 blot after the PDE2A2 modulation.

We thank this reviewer for suggesting this important experiment. We now include data showing that inhibition of PDE2A reduces the CCCP-induced accumulation of PINK1 both in untreated and, more significantly, in ISO treated cells (new Fig 7A-B). We also show that displacement of endogenous PDE2A2 with a catalytically inactive mutant

(PDE2A2DN) results in reduced accumulation of PINK1 (new Fig 7C-D), confirming the involvement of this isoform in the regulation of PINK1 stability. In addition, we now provide evidence that PDE2A inhibition also reduces PINK1 levels in white adipocytes treated with isoproterenol (Fig 8A), indicating that this pathway also operates in this cell type.

Minor points

1. In Fig. 3A, it is better to mention that the effect of BAY only can be observed in the lower panel (long exposure) of phospho-Mic60 blot.

Text has been amended as suggested.

2. In Fig. 3D and H, why the phospho-Mic60 blot shows ladder bands even in IP blot? At least, it is better to insert the molecular weight on this blot to show that the arrow head precisely indicates the immunoprecipitated Mic60.

There is obviously some contamination of our IP fractions with phosphorylated proteins other than MIC60. However, we can unequivocally identify the band that corresponds to phosphorylated MIC60 by overlay with the MIC60-FLAG signal. We agree with the reviewer that having molecular weight markers as a reference would help the reader to interpret the results and we have modified the images accordingly.

Reviewer #2:

Major points

1. The link between the MIC60/PDE2A2/PKA axis and mitophagy followed closely with work done by Akabane et al. published several years ago (PMID 27153535) showing that PKA phosphorylates MIC60 and inhibits mitophagy. While it was difficult for me to see what was gained mechanistically by further implicating PDE2A2 in this process, the authors seemingly ignored completely the main finding of that paper: that phosphorylation of MIC60 by PKA prevents PINK1 accumulation on mitochondria. The authors should expectedly see a similar modulation of PINK1 via genetic/pharmacological inhibition of PDE2A2. Is this so?

We would like to respectfully point out that the identification of PDE2A2 as a regulator of PKA-dependent phosphorylation of MIC60 and mitophagy significantly advances our mechanistic understanding of mitochondrial clearance. Our data demonstrate that PDE2A2 plays a role in mitophagy in cells that are otherwise untreated, emerging as a regulator of basal mitochondrial homeostasis. The exact function (or functions) of basal mitophagy remains unclear as are the cellular triggers and the signalling events driving this process. PDE2A2 activity can be potently activated by cGMP, a second messenger that has been linked to metabolic change and stress conditions in a number of tissues. In this context, PDE2A2 emerges as a key link between metabolism/stress and basal mitophagy. Further, PDE2A2 can be inhibited pharmacologically and can be activated

by treatments that elevate cGMP, thus our study reveals PDE2A as a potentially druggable target to treat disease conditions associated with either excess or deficient mitophagy.

We realise that in our original submission some of the above points were not explicitly discussed. We have now expanded a section of the discussion to include these points and we hope that the relevance of our findings in the mechanistic understanding of mitophagy emerges more clearly.

We completely agree with this reviewer that the effect of PDE2A2 manipulation on PINK1 accumulation on mitochondria is an important point to address. We have now performed additional experiments that confirm that pharmacological inhibition of PDE2A reduces the accumulation of PINK1, an effect that is significantly enhanced in the presence of isoproterenol, and that this mechanism also operates in adipocytes. In addition, we show that displacement of endogenous PDE2A2 from mitochondria via overexpression of a catalytically inactive mutant of this isoform results in greatly reduced accumulation of PINK1. These new data are shown in the revised manuscript in the new Fig 7 and Fig 8A.

2. It was difficult for me to appreciate the data concerning fat browning (Fig. 6F and G). These data immediately follow studies on mitophagy (i.e. mitochondrial degradation). It is my understanding that the authors are suggesting that PDE2A2 facilitates mitochondrial turnover via autophagy, and this may explain the increased mitochondrial load of brown fat vs. white adipocytes upon BAY-60 treatment. The authors are however measuring mRNA, implying that mitochondrial biogenesis is in fact increased. Additionally, does PDE2A2 inhibition also increase mitochondrial biogenesis in the cell culture models in which mitophagy is assessed?

Our data suggest that in white adipocytes, PDE2A2 regulates mitochondrial homeostasis in beige cells through mitophagy. Inhibition of PDE2A shifts the white adipocytes towards a browner phenotype. By inhibiting the PDE2A-dependent mitophagy, we are not only maintaining the beige phenotype induced by NE, but we increase the mitochondrial load as physiological mitochondrial clearance is attenuated. We appreciate the point raised by this reviewer that measuring mRNA of nucleus-encoded mitochondrial genes may be misleading as it may indicate increased mitochondria biogenesis rather than reduced mitophagy. However, we would argue that increased transcription of nuclear genes is required to maintain a larger pool of mitochondria resulting from reduced mitophagy. In any case, to address this point and to more directly assess the size of the mitochondrial pool we now include quantification of expression of genes encoded by mitochondrial DNA, normalised for the expression of a housekeeping gene (data shown in Fig 8D).

We have not assessed mitochondrial load in other cell models in the current study. This is an important aspect and will be the focus of follow up investigations.

3. Dissection of the MIC60-PDE2A2 relationship – arguably the crux of the manuscript – was fairly straightforward and logically approached. The authors ultimately suggest via

truncation experiments that the GAFB domain targeted to the IMS is sufficient to bind MICOS. Determining whether or not a chimera of GAFB and an IMS-targeting sequence of another protein entirely (and not the PDE2A2 N-terminus) would demonstrably show that this domain is sufficient to bind MIC60.

We thank this reviewer for suggesting this experiment to conclusively demonstrate that the GAF-B domain is sufficient to bind MIC60. As described in the revised manuscript we have modified the N-terminal+GAF-B-RFP construct by substituting the mitochondrial targeting sequence from PDE2A2 with the mitochondrial targeting sequence from the MIS protein Diablo/Smac. We show by imaging (Fig S3E) and by western blot (Fig S3F) that this new chimera correctly localises to the mitochondria and that it can effectively be immunoprecipitated by MIC60-FLAG pull-down (Fig 2C)

4. Does overexpressing PDE2A2 suppress MIC60 phosphorylation (i.e. in relation to Fig. 3)?

Again, we thank this reviewer for suggesting this experiment. We now include data (Fig S4 C-D) showing that overexpression of PDE2A2 completely suppresses MIC60 phosphorylation induced by forskolin treatment.

5. The authors use the term “energized mitochondria” several times throughout the text to mean experiments performed in the absence of uncoupler. It is very likely that the specific organelles in question are depolarized, however, as a small number of mitochondria may become depolarized in a cell at a given time. This should be made clearer in the text. If the authors truly believe that these parkin-positive mitochondria have a membrane potential, they should demonstrate this by using a dye such as TMRM.

We apologise for misusing the term ‘energised mitochondria’ to mean mitochondria in cells that are not treated with uncoupler. We have now amended the text accordingly.

Minor points

1. It would benefit the reader by initially explaining the function of each GAF domain.

Details on the function of the GAF domains have now been included in the introduction

2. MIC60 levels are not visible in the input for the IPs in Fig. 3. Can the authors provide longer exposures?

MIC60 is not visible in the input in Fig 3 due to the low amount of protein loaded to balance the signal from the IP. Unfortunately, we do not have images of these membranes taken with longer exposure.

3. The authors should explicitly state that the antibody used in Fig. 3 is against PKA phosphorylation.

This was indicated in the original submission in the figure (labelled as: PKA sub). We now more explicitly state this in the legend to Fig 3

4. Can the authors provide information in Materials and Methods for how blots were quantified? This is particularly important in the case of IPs.

Information is now included

5. The authors should show accompanying microscopy images for the data quantified in Fig. 5A-F and Fig. 6A-E. Moreover, are the cells in Fig. 5F treated with CCCP? If not, is it not surprising that >60% of cells overexpressing MIC60 show parkin translocation? Is this translocation only partial?

We now provide images representative for the most important experiments shown in Fig 4, 5 and 6. The cells shown in Fig 5F (now 5K in the revised manuscript) were not treated with CCCP. Overexpression of MIC60 or of its S528A mutant is sufficient to promote Parkin recruitment likely via increased stabilisation of PINK1 at the membrane. We expect transfection efficiency to be high in MEFs, so it is not surprising to find that a high proportion of cells show Parkin recruitment. The amount of Parkin recruited to the mitochondria in these cells is indeed partial.

Reviewer #3:

1- The authors claim that they identified the interactome of PDE2A2 specifically, and not other members of PDE2A. However, it is not clear to this reviewer why the authors used Elongation factor thermo stable protein (EF-T) from *E. coli* as negative control for the strep-tag pull down, rather than using the sequence encoding other members of PDE2A, the negative control of the pull down should have been sequence encoding PDE2A member other than PDE2A2. As the authors already indicated in the context of their co-IP's experiments, some mitochondrial substrates are strictly PDE2A-isoform dependent and hence distinguishing the specific substrates for each isoform might be feasible.

It is certainly feasible and important to study the specific interactome of PDE2A1 and PDE2A3 to complement the current findings and these are experiments that we are planning to do as a follow up to this study. Here, the focus was on PDE2A2 and the use of proteomics to dissect the subcellular localisation of this isoform, stemming from previous studies showing that PDE2A2 localises to different sub-mitochondrial compartments as well as extramitochondrial sites. In that context, EF-T was used as a control for tag overexpression.

2- I'm a little confused by the mass spec results. In Fig. 1A, MIC60, MIC19 and SAMM50 appear prominent. However, when I scanned the supplemental data 1, I could not even find MIC60 or MIC19 among the list. SAMM50 was not even among the top 50

hits. While it makes sense to focus on the MICOS complex biologically, it makes one wonder about the rationale or usefulness of the mass spec data in the logic of this story.

We thank the reviewer for this observation. Indeed, the names indicated in the plot are not in the table because the table reports the official gene names, while the plot reports a commonly used Synonym. We have now addressed this issue by including in Fig 1A the official gene names alongside the synonym.

Regarding SAMM50, its position in the volcano plot depends on the enrichment of the protein in the IP (average of the replicate experiments) and the reproducibility of the enrichment between replicates. Conversely, the order of the protein in the original Supplementary Data was based on the enrichment (ratio PDE2/CTL) calculated in the first experiment; this means that the order of the proteins in the supplementary table was not a ranking. We now provide a revised table where we have ranked the proteins differently to better mirror the volcano plot. We put at the top those proteins with a minimum PDE2/CTL ratio of 2 and we further rank them based on the calculated p-value. IMMT is now ranked 1st, CHCHD3 20th and SAMM50 24th.

3- I have the same comment for supplemental data 2. It seems a bit construed to say that mitochondrial membrane was selectively enriched in the GO analysis when it was ranked 21th out of 26 pathways.

Also for this table, in the original submission, we simply listed the GOCC categories with no specific ranking. When ordering them by FDR, “Mitochondrial membrane” is ranked 10th. Of note, the top ranked category, which is “Integral to membrane”, contains most of the proteins annotated also in “Mitochondrial membrane”. Hence, this strongly justifies our choice of focusing on mitochondrial membrane proteins. We now provide a revised Supplementary Data 2 with GO categories ranked by FDR.

4- P15, Lines380-381, “overall the above data support a role of PDE2A2 in regulation of” this statement is not strictly true, because the authors in this part investigated the role of PDE2A2 on regulation of PKA-mediated phosphorylation of MIC60 using BAY60-7550, an inhibitor for PDE2 members and not sole inhibitor for PDE2A2. Further, the genetic results obtained from PDE2A knockout that is devoid of the all the three isoforms of PDE2A (PDE2A1, PDEA2 and PDEA3). Thus, given this context of results, it is hard to dissect the distinct and strict role of PDE2A2 in regulation of PKA-phosphorylation of MIC60 without taking in consideration the role of other PDE2A’s. In vitro kinase assays and/or overexpression of different forms PDE2A2 (active and kinase dead) might help in solidifying this part. At the very least, this point should be discussed.

We agree with the reviewer that the set of experiments summarised in Fig 3 do not allow to dissect which isoform is involved and we have now reworded the conclusive statement in the relevant paragraph to say ‘Overall, the above data support a role of PDE2A in the regulation of PKA-mediated phosphorylation of MIC60’. However, we provide evidence in other sections of the results that supports a specific role of PDE2A2 in the regulation of mitophagy. This includes: i) only PDE2A2 interacts with MIC60 (Fig 1G); ii) in a new set of data we show that the effects of PDE2A2 overexpression on

mitophagy as detected by the mito-QC reporter, are not recapitulated by PDE2A1 or PDE2A3 overexpression (new Fig 4I-J). We therefore believe we can confidently conclude that it is the 2A2 isoform that is involved in the regulation of mitophagy.

5- While the authors indicated that PDE2A inhibition can increase the PKA-dependent Phosphorylation of MIC60, reduce parkin recruitment to the mitochondria and attenuate mitophagy, a number of molecular aspects are not clear, for instance, how local cAMP modulation, via modulating levels of PDE2A and MIC60 phosphorylation, signals to alter parkin recruitment and mitophagy, this should be at least pointed out clearly in the discussion section.

We apologise for not making this point sufficiently clear. PKA-dependent phosphorylation of MIC60 has been shown to destabilise the interaction of MIC60 with PINK1 at the mitochondrial membrane resulting in rapid degradation of PINK1 by the proteasome and consequently, reduced parkin recruitment to the mitochondria (Akabane, S. *et al. Mol. Cell* **62**, 371–384 (2016)). Our model would therefore predict that inhibition of PDE2A2 results in local increase of cAMP and PKA activation in the MIS, destabilization of the MIC60/PINK1 interaction and reduced Parkin recruitment. We have now included a more explicit description of this mechanism in the revised text and performed additional experiment to address the role of PINK1 (see point below).

6- While the authors revealed the role of inhibition of PDE2A on PKA-mediated phosphorylation of MIC60, parkin recruitment and mitophagy, the role of PINK1 kinase, a crucial factor in PARKIN-dependent mitophagy has not been addressed. PINK1 has been shown to phosphorylate MIC60 (Please check: PMID: 29456190) and this paper should be discussed. Whether PINK1 plays a role in the reported PDE2A-mediated regulation of mitophagy is not yet clear. It may be informative to investigate or at least discuss how the modulation of PDE2A could affect PINK1 and vice versa.

We completely agree with this reviewer that it was important to investigate the involvement of PINK1 in the process regulated by PDE2A2. To address this point, we have performed a new set of experiments that show that PDE2A inhibition in cells treated with CCCP results in reduced accumulation of PINK1 (new Fig 7A-B). In addition, we show that displacement of endogenous PDE2A2 with a catalytically inactive mutant equally results in reduced PINK1 accumulation (Fig 7C-D). As we now discuss in the revised manuscript, our findings are compatible with a model where inhibition of PDE2A2 results in local increase of cAMP, PKA-mediated phosphorylation of MIC60 and consequent destabilization of the interaction of MIC60 with PINK1 at the mitochondrial membrane, resulting in rapid degradation of PINK1 by the proteasome and consequently, reduced parkin recruitment to the mitochondria. We now include the suggested reference in the revised discussion where we mention the fact that MIC60 can be phosphorylated by different kinases, including PINK1, leading to different effects.

7- Finally, the experiments on adipocyte browning are very superficial and premature. There is no clear link to MIC60 phosphorylation or to the PINK1/Parkin pathway to

justify its inclusion in the abstract and discussion. Unless, these connections can be made, this part should be de-emphasized or ideally removed.

We have now significantly expanded the data on the role of PDE2A inhibition on adipocyte browning. As shown in the new Fig 8A-B, we demonstrate a link between inhibition of PDE2A and accumulation of PINK1 in primary white adipocytes. We further demonstrate that PDE2A inhibition increases the mitochondria pool by showing enhanced expression of mitochondria-encoded genes (new Fig 8D) and enhanced mitochondria respiration (new Fig 8E). In addition, we show that inhibition of PDE2A potentiates the effect of norepinephrine on lipid droplet size (new Fig 8F). Although we recognise that further studies will be necessary to fully dissect the role of PDE2A inhibition in fat tissue metabolism and possible implications, we believe the data presented here provide novel and solid evidence that PDE2A is relevant in this context.

8- Overlap of mitochondrial markers with LC3 is not a great way to quantify mitophagy. Have the authors considered using more quantitative approaches such as mito-Keima or mitoQC reporter systems? I don't think it is essential for this paper but I would highly recommend it if the authors want to pursue this type of work.

Following the suggestion by this reviewer, our revised manuscript now includes a series of experiments where we have evaluated mitophagy using the mito-QC reporter system. Specifically, using this approach we were able to confirm that overexpression of PDE2A2, but not of PDE2A1 or PDE2A3, increases basal mitophagy in MEFs in the absence of CCCP (Fig 4I-J) and that inhibition of PDE2A attenuates mitophagy induced by CCCP treatment (Fig 5D-E) as well as basal mitophagy (Fig 6E-F), an effect that is not recapitulated by PDE3 inhibition. We believe that these new data significantly strengthen our conclusion that PDE2A2 is involved in the regulation of mitophagy.

Reviewers' comments:

Reviewer #1 (Remarks to the Author):

The authors added several new data based on this and other reviewers' suggestions, but several points are still premature. My major concern is that newly added representative image data of Parkin recruitment (Fig. 5b, for example) are really hard to see. I cannot judge whether the bar graph in Fig. 5a really reflects their observation. mtQC data also contains severe problems. For example, in Fig. 5d, we cannot see any red dots (which represent mitophagy) even in the CCCP-treated condition, but the authors count something and show the bar graph in Fig. 6e. These points raise a question about the reliability of every bar graph data set. It seems the authors do not express Parkin exogenously in MEF cells for mtQC experiment, which may make them hard to observe mitophagy in MEF cells. To improve this point, it is recommended to check the expression level of endogenous Parkin in their mtQC-stable MEF cells, and express Parkin exogenously. Or, they can optimize the concentration of CCCP and the duration of CCCP treatment up to 24 hours. The authors added PINK1 immunoblotting data in Fig. 7a, but compared to the drastic effect of Forskolin on PINK1 accumulation, the effect of BAY was not striking to me under CCCP-treated condition (I agree that it seems there was some effect under ISO-treated condition). It is consistent with its little effect on Mic60 phosphorylation. In white adipocytes (newly added in Fig. 8a), I could not see any difference in PINK1 accumulation, although they showed some statistically significant difference in the bar graph. It is better to check whether indicated PINK1 band is a full-length form (Fig. 8a). The authors can easily test this using MG132, a proteasome inhibitor which accumulates a cleaved-form of PINK1 (around 50 kDa)(Fig. 8a). Additionally, PINK1 knockdown would be better to check whether these bands actually represent endogenous PINK1(Fig. 8a). Also, if they want to connect their observation with being, more direct approach including PINK1 knockdown etc. will be necessary(Fig. 8a).

Reviewer #2 (Remarks to the Author):

The revised manuscript addresses the points raised upon initial submission. Specifically, the authors were able to address my points mostly through elegant experimentation. Additionally, the new PINK1 data really ties together the mitophagy work done in heterologous cell culture and the data on adipocytes. Congratulations to the authors on their solid work and contribution to the field!

Reviewer #3 (Remarks to the Author):

This is a revised manuscript. The authors have addressed my comments with new data and revised text. There are just few comments to enhance the work.

1- In Fig 7a legend, it is not clear for how long the authors treated the cells with CCCP. Whether the authors used crude whole cell extracts or mitochondrial fractions is not also clear. This should be clarified.

2- This reviewer doesn't fully appreciate what is meant by the author's statement "we show that displacement of endogenous PDE2A2 from mitochondria via overexpression of a catalytically inactive mutant of this isoform results in greatly reduced accumulation of PINK1". It is difficult to appreciate the inactivation mutant used particularly when the authors didn't mention, in the current manuscript, the details regarding the position of inactivation mutant (PDE2A2DN), this need to be indicated in the current manuscript. How this mutation was generated is not clear in the text. This should be clarified at least in the methods section.

3- The authors overuse the term "degradation" when they refer to pink1 steady-state levels reduction in different places (For instance, "inhibition of PDE2A potentiates the effect of ISO in

promoting PINK1 degradation"). Strictly speaking, this manuscript doesn't have a single proper protein degradation assay like pulse-chase assays, ..etc. Thus, the authors can't really claim that certain condition promotes degradation more than other (Had the authors perform half-life measurements for pink1 under both conditions to know whether it is promoting degradation or not). The authors are advised to use "reduction of PINK1-levels" when referring to their data regarding the mitigation of PINK1 steady state levels. Surely, pink1 is known to be regulated by protein degradation. Nevertheless, there are no data in the current manuscript to be sure that the effect seen is, in fact, due to degradation rather than limited proteolysis "restricted proteolysis". In other words, the reduced intensity of the PINK1 band, under certian condition, does not necessarily mean that it is degraded.

4- Some controls are missing (figure 7C), like the expression of PDE2A2-DN-RFP inactive mutant in absence of CCCP.

A point-by-point rebuttal (in blue) to the issues raised by the reviewers (unedited, in black) is presented below.

Reviewer #1

The authors added several new data based on this and other reviewers' suggestions, but several points are still premature. My major concern is that newly added representative image data of Parkin recruitment (Fig. 5b, for example) are really hard to see. I cannot judge whether the bar graph in Fig. 5a really reflects their observation. mtQC data also contains severe problems. For example, in Fig. 5d, we cannot see any red dots (which represent mitophagy) even in the CCCP-treated condition, but the authors count something and show the bar graph in Fig. 6e. These points raise a question about the reliability of every bar graph data set.

We apologise for the poor quality of the images of cells showing Parkin recruitment and mito-QC puncta (e.g. Fig 5 and Fig 6) as they appear in the pdf file. In the original high-resolution images these details are very clear and, we believe, entirely convincing. To allow the reviewers to see the high-resolution images we have submitted them as separate .tiff files. We are confident that this will resolve this issue.

It seems the authors do not express Parkin exogenously in MEF cells for mtQC experiment, which may make them hard to observe mitophagy in MEF cells. To improve this point, it is recommended to check the expression level of endogenous Parkin in their mtQC-stable MEF cells, and express Parkin exogenously. Or, they can optimize the concentration of CCCP and the duration of CCCP treatment up to 24 hours.

We agree with this reviewer that overexpression of Parkin and extended treatment with CCCP may increase the amount of Parkin recruited to the mitochondria and the mito-QC signal. However, we preferred to avoid further cell manipulation to minimise confounding effects that may be due to protein overexpression or prolonged toxic effect from the uncoupler. As mentioned above, we believe that the effect of PDE2A2 on Parkin recruitment and the mito-QC signal can be appreciated very clearly in the original high-resolution images, so in our opinion the suggested additional manipulations are not necessary.

The authors added PINK1 immunoblotting data in Fig. 7a, but compared to the drastic effect of Forskolin on PINK1 accumulation, the effect of BAY was not striking to me under CCCP-treated condition (I agree that it seems there was some effect under ISO-treated condition). It is consistent with its little effect on Mic60 phosphorylation.

Indeed, the effect of forskolin on MIC60 phosphorylation and Pink1 protein levels is significantly larger than observed on PDE2A inhibition. This is expected though as at 25 μ M, the concentration of forskolin used in this experiment, all the adenylyl cyclases in the cells are maximally activated. This results in an increase in cAMP concentration that is much larger than the increase in cAMP that results from its reduced degradation via PDE2A inhibition, as the cyclases are working at their basal level in this setting. Consistently, the effect of Bay becomes more evident on ISO treatment: in this scenario, ISO activates the cyclases coupled with beta-adrenergic receptors and inhibition of PDE2A results in a more obvious increase in cAMP and a significant reduction of Parkin levels.

In white adipocytes (newly added in Fig. 8a), I could not see any difference in PINK1 accumulation, although they showed some statistically significant difference in the bar graph. It is better to check

whether indicated PINK1 band is a full-length form (Fig. 8a). The authors can easily test this using MG132, a proteasome inhibitor which accumulates a cleaved-form of PINK1 (around 50 kDa)(Fig. 8a). Additionally, PINK1 knockdown would be better to check whether these bands actually represent endogenous PINK1(Fig. 8a).

We realise that the differences in band intensity in the western blot shown in Fig 8a are small. This is, at least in part, because loading is not completely even across samples (as shown by the calnexin signal). However, densitometric analysis of the Pink1 signal and normalization to total protein (calnexin signal) in five independent experiments clearly shows differences as illustrated in the summary graph in Fig 8a. Based on its molecular weight and the fact that no cleaved form at lower molecular weight is detected on the membrane, we believe that the band shown in Fig 8a is full length Pink1 (the image of the full membrane used to generate Fig 8A is shown below). As there is no confounding signal on the western blot and the antibody used here has been extensively used in the literature (over 60 publications since 2013) and is considered to be reliable, we believe we are looking at endogenous Pink1. In particular, our findings are in agreement with published data using the same antibody (see for example Fig 2 in Akabane et al, Molecular Cell 2016) showing a single band for endogenous Pink1 at around 58 kDa

Also, if they want to connect their observation with beiging, more direct approach including PINK1 knockdown etc. will be necessary (Fig. 8a).

We thank this reviewer for the suggestion. We believe that our finding that PDE2A2, an easily druggable target, is involved in beiging is very exciting and we agree that it deserves more thorough investigations. However, to achieve significant further understanding, the work required would need to extend beyond PINK1 knockdown analysis. While this is beyond the scope of the current study, we are planning to undertake this in the future.

Reviewer #2

The revised manuscript addresses the points raised upon initial submission. Specifically, the authors were able to address my points mostly through elegant experimentation. Additionally, the new PINK1 data really ties together the mitophagy work done in heterologous cell culture and the data on adipocytes. Congratulations to the authors on their solid work and contribution to the field!

We thank this reviewer for the kind words and recognition of our efforts!

Reviewer #3

This is a revised manuscript. The authors have addressed my comments with new data and revised text. There are just few comments to enhance the work.

1- In Fig 7a legend, it is not clear for how long the authors treated the cells with CCCP. Whether the authors used crude whole cell extracts or mitochondrial fractions is not also clear. This should be clarified.

We thank this reviewer for pointing this out. Clarification is now included in the legend to Fig 7a

2- This reviewer doesn't fully appreciate what is meant by the author's statement "we show that displacement of endogenous PDE2A2 from mitochondria via overexpression of a catalytically inactive mutant of this isoform results in greatly reduced accumulation of PINK1". It is difficult to appreciate the inactivation mutant used particularly when the authors didn't mention, in the current manuscript, the details regarding the position of inactivation mutant (PDE2A2DN), this need to be indicated in the current manuscript. How this mutation was generated is not clear in the text. This should be clarified at least in the methods section.

We apologise for this omission. Full detail of the mutations introduced and reference to the original publication is now included on p 17 in the revised manuscript.

3- The authors overuse the term "degradation" when they refer to pink1 steady-state levels reduction in different places (For instance, "inhibition of PDE2A potentiates the effect of ISO in promoting PINK1 degradation"). Strictly speaking, this manuscript doesn't have a single proper protein degradation assay like pulse-chase assays, ..etc. Thus, the authors can't really claim that certain condition promotes degradation more than other (Had the authors perform half-life measurements for pink1 under both conditions to know whether it is promoting degradation or not). The authors are advised to use "reduction of PINK1-levels" when referring to their data regarding the mitigation of PINK1 steady state levels. Surely, pink1 is known to be regulated by protein degradation. Nevertheless, there are no data in the current manuscript to be sure that the effect seen is, in fact, due to degradation rather than limited proteolysis "restricted proteolysis". In other words, the reduced intensity of the PINK1 band, under certain condition, does not necessarily mean that it is degraded.

The text in the revised manuscript has been amended accordingly

4- Some controls are missing (figure 7C), like the expression of PDE2A2-DN-RFP inactive mutant in absence of CCCP.

Expression of the catalytically inactive PDE2A2-DN construct reduces the level of PINK1 in cells treated with CCCP (as shown in the last lane in Fig 7C). However, as in our experimental conditions, PINK1 is already hardly detectable in the absence of CCCP treatment (first lane in Fig 7C) we would not expect to be able to detect any significant difference on expression of PDE2A2-DN. This is also confirmed by data shown in Fig 4F, where, in the absence of CCCP treatment, expression of PDE2A2-DN has no effect on Parkin recruitment, which occurs downstream of Pink1. For these reasons we believe that the additional control suggested here would not provide a meaningful result.

REVIEWERS' COMMENTS:

Reviewer #1 (Remarks to the Author):

Comments to the authors

Updated high-resolution figures support the authors' conclusion.
I appreciate their efforts to improve this point.